# Evaluating Prophylactic Effect of Bovine Colostrum on Intestinal Barrier Function in Zonulin Transgenic Mice: A Transcriptomic Study

**DOI:** 10.3390/ijms241914730

**Published:** 2023-09-29

**Authors:** Birna Asbjornsdottir, Snaevar Sigurdsson, Alba Miranda-Ribera, Maria Fiorentino, Takumi Konno, Jinggang Lan, Larus S. Gudmundsson, Magnus Gottfredsson, Bertrand Lauth, Bryndis Eva Birgisdottir, Alessio Fasano

**Affiliations:** 1Department of Pediatric Gastroenterology and Nutrition, Mucosal Immunology and Biology Research Center, Massachusetts General Hospital, Boston, MA 02114, USA; basbjornsdottir@mgh.harvard.edu (B.A.); rosaria_f@hotmail.com (M.F.); tkonno@sapmed.ac.jp (T.K.); jlan1@mgh.harvard.edu (J.L.); 2School of Health Sciences, Faculty of Medicine, University of Iceland, 102 Reykjavik, Icelandmagnusgo@landspitali.is (M.G.);; 3Unit for Nutrition Research, Landspitali University Hospital, Faculty of Food Science and Nutrition, University of Iceland, 102 Reykjavik, Iceland; 4Biomedical Center, University of Iceland, 102 Reykjavik, Iceland; 5Department of Cell Science, Research Institute for Frontier Medicine, Sapporo Medical University School of Medicine, Sapporo 060-8556, Japan; 6School of Health Sciences, Faculty of Pharmaceutical Sciences, University of Iceland, 102 Reykjavik, Iceland; 7Department of Scientific Affairs, Landspitali University Hospital, 102 Reykjavik, Iceland; 8Department of Infectious Diseases, Landspitali University Hospital, 102 Reykjavik, Iceland; 9Department of Child and Adolescent Psychiatry, Landspitali University Hospital, 102 Reykjavik, Iceland; 10Department of Pediatrics, Harvard Medical School, Harvard University, Boston, MA 02138, USA

**Keywords:** cingulin, neuroinflammation, claudin, dysbiosis, gut permeability, occludin, NF-kB signaling pathway, tight junctions, mRNA sequencing

## Abstract

The intestinal barrier comprises a single layer of epithelial cells tightly joined to form a physical barrier. Disruption or compromise of the intestinal barrier can lead to the inadvertent activation of immune cells, potentially causing an increased risk of chronic inflammation in various tissues. Recent research has suggested that specific dietary components may influence the function of the intestinal barrier, potentially offering a means to prevent or mitigate inflammatory disorders. However, the precise mechanism underlying these effects remains unclear. Bovine colostrum (BC), the first milk from cows after calving, is a natural source of nutrients with immunomodulatory, anti-inflammatory, and gut-barrier fortifying properties. This novel study sought to investigate the transcriptome in BC-treated Zonulin transgenic mice (Ztm), characterized by dysbiotic microbiota, intestinal hyperpermeability, and mild hyperactivity, applying RNA sequencing. Seventy-five tissue samples from the duodenum, colon, and brain of Ztm and wild-type (WT) mice were dissected, processed, and RNA sequenced. The expression profiles were analyzed and integrated to identify differentially expressed genes (DEGs) and differentially expressed transcripts (DETs). These were then further examined using bioinformatics tools. RNA-seq analysis identified 1298 DEGs and 20,952 DETs in the paired (Ztm treatment vs. Ztm control) and reference (WT controls) groups. Of these, 733 DEGs and 10,476 DETs were upregulated, while 565 DEGs and 6097 DETs were downregulated. BC-treated Ztm female mice showed significant upregulation of cingulin (*Cgn*) and claudin 12 (*Cldn12*) duodenum and protein interactions, as well as molecular pathways and interactions pertaining to tight junctions, while BC-treated Ztm males displayed an upregulation of transcripts like occludin (*Ocln*) and Rho/Rac guanine nucleotide exchange factor 2 (*Arhgf2*) and cellular structures and interfaces, protein–protein interactions, and organization and response mechanisms. This comprehensive analysis reveals the influence of BC treatment on tight junctions (TJs) and Nuclear factor kappa-light-chain-enhancer of activated B cells (NF-kB) signaling pathway gene expressions. The present study is the first to analyze intestinal and brain samples from BC-treated Ztm mice applying high-throughput RNA sequencing. This study revealed molecular interaction in intestinal barrier function and identified hub genes and their functional pathways and biological processes in response to BC treatment in Ztm mice. Further research is needed to validate these findings and explore their implications for dietary interventions aimed at improving intestinal barrier integrity and function. The MGH Institutional Animal Care and Use Committee authorized the animal study (2013N000013).

## 1. Introduction

In recent years, the pivotal role of the intestinal barrier in maintaining overall health has gained increased attention in the scientific community [1,2,3]. The intestinal barrier serves as a critical defense line, segregating the internal milieu from the external environment laden with potential pathogens, toxins, and allergens [4]. Its compromise, particularly in response to inflammation or infection, can lead to increased paracellular movement of molecules and ions, leading to potentially deleterious effects. A disrupted intestinal barrier can activate immune cells, trigger an inflammatory response, and facilitate the entry of inflammatory molecules into the circulation, thereby contributing to non-intestinal diseases, including systemic and neuroinflammation [5,6,7].

Integral to the intestinal barrier’s integrity are tight junctions (TJs). TJs, intercellular adhesion complexes, encircle the apical part of the epithelial cell’s lateral membrane, creating a continuous belt-like structure [8,9]. TJs consist of transmembrane proteins, such as claudin and occludin, junctional adhesion molecule-A (JAM-A), and intracellular plaque proteins, such as zonula occludens (ZO) [10]. These proteins, amongst others, work together to regulate the paracellular movement of molecules and ions and to maintain the physical barrier. Transmembrane proteins form strands between adjacent cells, while intracellular plaque proteins provide a structural scaffold to support the TJ structure. The cytoplasmic domains of the transmembrane proteins are associated with the cytoplasmic domains of the intracellular plaque proteins through protein–protein interactions. These interactions are vital for maintaining the integrity of the TJ structure [10].

Dietary factors and nutrition play a critical role in regulating the tight junction barrier and paracellular permeability [11]. Among these factors, bovine colostrum (BC), a nutrient-rich fluid produced by cows post-calving, is known for its immunomodulatory, anti-inflammatory, and gut-barrier-fortifying properties [12]. These effects are primarily attributable to its bioactive components, such as immunoglobulins, growth factors, lactoferrin, alpha-lactalbumin, and oligosaccharides, which collectively promote gut health, immune function, and tissue repair [13,14,15]. Recently, micro RNAs (miRNA) were found to be part of this bioactivity, possibly acting as key regulators of diverse biological and developmental processes [16]. miRNAs may influence intestinal development and immunity, driving cell viability, proliferation, and stem cell activity in the intestinal epithelium [17]. Moreover, the synergistic action of these bioactive components could amplify the beneficial effects of BC, providing an integrated response that might be more potent than the sum of its individual elements [18]. However, research on BC’s application to the intestinal barrier remains relatively sparse.

In this light, the present study leveraged the Ztm model, characterized by a predisposition to inflammation, including neuroinflammation, due to a dysbiotic microbial community and increased trafficking of pro-inflammatory bacterial species and their products into the systemic circulation [19,20]. Utilizing high-throughput RNA sequencing on tissues from Ztm and WT mice, our research probes BC’s protective effects on intestinal barrier integrity and the transcriptomic responses in both the digestive tract (i.e., small intestine and colon) and brain. Our findings illuminate molecular interactions, pathways, and processes dictating intestinal barrier integrity in response to BC. This novel understanding may unveil potential therapeutic avenues to enhance gut barrier function, presenting preventive strategies against inflammation and its associated ailments.

## 2. Results

A total of 75 tissue samples were sequenced, comprising 25 duodenum samples, 25 colon samples, and 25 brain samples sourced from Ztm (*n* = 12) and WT (*n* = 13) mice. After discarding low-quality reads, reads containing adapters, and those with poly-N sequences, a total of 510,554,764 clean reads were obtained.

To investigate the effects of colostrum intervention on gene and transcript expression, we performed a comparative analysis of mRNA expression levels in Ztm and WT control mice at baseline and between the treatment and control groups. Specifically, this analysis focused on identifying differentially expressed genes (DEGs) and differentially expressed transcripts (DETs) across these conditions.

By assessing baseline differences, we aimed to understand the potential influence of genotype on gene expression patterns and to establish a reference point for subsequent comparisons. Furthermore, analyzing the data focusing on the treatment and control groups within the Ztm mice enabled us to discern the specific effects of the intervention on gene expression in the context of the Ztm model.

All comparisons were performed and analyzed separately to account for gender-specific responses, aiming to capture any gender-specific variations in gene expression patterns and identify potential gender-dependent effects of the colostrum intervention.

### 2.1. Differential Expression of Genes and Transcripts across Study Groups

A comprehensive transcriptomic analysis was performed on all samples. RNA-seq analysis was used to identify 1298 differentially expressed genes (DEGs) and 20,952 differentially expressed transcripts (DETs) in the paired (Ztm treatment vs. Ztm control) and reference (WT controls) groups with adjusted *p*-value (q ≤ 0.05) and|log2 FC| ≥ 1. In the current study, 733 DEGs and 10,476 DETs were upregulated, while 565 DEGs and 6097 DETs were downregulated. Table 1 provides an extensive breakdown of these DEGs and DETs, segmenting data according to specific groups and organ types.

#### 2.1.1. Gene and Transcript Alterations in BC-Treated Ztm Mice: A Duodenal, Colonic, and Cerebral Perspective Compared to Control Mice

In the BC-treated Ztm female group, 82 genes were differentially expressed, i.e., 46 upregulated and 36 downregulated, compared with control mice. In the duodenum, 46 genes were upregulated, and 32 were downregulated, whereas in the colon, no genes were upregulated, but 4 genes were downregulated. However, no differentially expressed genes were found in the brain. The number of upregulated transcripts in the duodenum, colon, and brain were 1180, 416, and 494, while 11, 76, and 126 transcripts were downregulated, respectively.

In the BC-treated Ztm male group, 439 genes were found to be differentially expressed, 242 genes upregulated, and 197 genes downregulated, compared with the control group. In the duodenum, 13 genes were upregulated and 54 were downregulated; in the colon, 228 genes were upregulated and 137 were downregulated. In the brain, one gene was upregulated, and six were downregulated. There were 359, 604, and 606 upregulated transcripts in the duodenum, colon, and brain and 1402, 306, and 220 downregulated transcripts, respectively.

#### 2.1.2. Gender Differences and Tissue-Specific Gene Expression in Ztm Mice: A Comparison with WT Reference Mice Using Differential Cluster Heat Maps

In the Ztm female control group, 61 genes were differentially expressed, i.e., 47 upregulated and 14 downregulated, compared with the WT female reference group. In the duodenum, 11 genes were upregulated and 5 were downregulated, whereas in the colon, 7 were upregulated and 3 were downregulated. However, 29 genes were found to be upregulated in the brain, and 6 were downregulated. The number of upregulated transcripts in the duodenum, colon, and brain were 86, 238, and 1123, with 470, 206, and 210 downregulated, respectively.

In the Ztm male control group, 101 genes were found to be differentially expressed, 72 genes upregulated, and 29 genes downregulated, compared with the WT male reference group. In the duodenum, 20 genes were upregulated and 8 were downregulated, and in the colon, 43 were upregulated and 20 were downregulated. In the brain, nine genes were upregulated and one was downregulated. The upregulated transcripts in the duodenum, colon, and brain numbered 982, 225, and 370, while 97, 465, and 931 transcripts were downregulated, respectively. In Figure 1, the DEGs are illustrated in expression heat maps (FPKM) and differential cluster heat maps (log2 FC) for duodenum and colon for Ztm female and male mice (BC and control) and WT female and male reference mice.

#### 2.1.3. Gender- and Organ-Specific Transcriptomic Responses to BC Treatment in Ztm Mice: An In-depth Look at Gene Expression Variances

In the BC-treated Ztm female group, a total of 124 genes were found to be differentially expressed, i.e., 68 genes upregulated and 56 genes downregulated when compared with the WT female reference group. In the duodenum, 58 genes were upregulated and 39 downregulated and, in the colon, 8 were upregulated and 11 were downregulated. In the brain, two genes were upregulated and six genes were downregulated. The upregulated transcripts in the duodenum, colon, and brain numbered 517, 575, and 1514, with 239, 121, and 42 downregulated, respectively.

In the BC-treated Ztm male group, 491 genes were differentially expressed, i.e., 258 genes were upregulated and 233 genes downregulated compared with the WT male reference group. In the duodenum, 209 genes were upregulated and 193 downregulated; in the colon, 48 were upregulated and 40 were downregulated. However, only one gene was upregulated in the brain, and none were downregulated. There were 527, 322, and 338 upregulated transcripts in the duodenum, colon, and brain, and 404, 380, and 391 downregulated transcripts, respectively.

Overall, there were substantial differences in gene expression changes between BC-treated Ztm mice, Ztm control mice, and WT reference mice across both sexes and different organs. Additionally, the number of genes with altered expression varied considerably between the organs, with the most prominent changes generally observed in the duodenum and colon. These observations suggest that BC treatment may cause organ-specific changes in gene expression in the Ztm model, and these effects can differ between females and males.

### 2.2. BC Treatment in Ztm Mice: Dissecting the Tissue-Specific Transcriptome and Genomic Response

In the BC-treated Ztm female vs. Ztm female control group, most gene expression differences were observed in the duodenum, with no significant gene changes in the brain and only downregulated DEGs in the colon. A large number of upregulated transcripts were found in the duodenum. The comparison between BC-treated Ztm males vs. Ztm male controls showed extensive changes, particularly in the duodenum and colon. Downregulated transcripts were particularly high in the duodenum.

In the Ztm female control group compared with the WT female reference group, relatively fewer changes were found in the duodenum and colon, with more significant gene and transcript changes in the brain, while within the Ztm male control vs. WT male reference group, differences were noted across all tissues. The most substantial gene regulation was in the colon, but the duodenum had more upregulated transcripts. The brain, however, showed more downregulated transcripts.

In the Ztm female BC-treated vs. WT female reference group, changes were noted across all tissues, with the brain showing the highest number of upregulated transcripts, whereas in the Ztm male BC-treated group vs. WT male reference group, the duodenum saw the most significant changes, with a high number of both upregulated and downregulated genes and transcripts.

In summary, BC treatment led to significant changes in the duodenum but relatively fewer changes in the colon and brain in Ztm female mice and significantly impacted gene expression and transcript levels in Ztm male mice, particularly in the duodenum and colon. It is evident that both genotype and treatment effects play roles in these organs. This underscores the importance of considering both genotype and treatment effects when analyzing gene and transcript expression dynamics.

Furthermore, transcript changes were far more numerous than gene changes, indicating that much of the difference may be due to alternative splicing or post-transcriptional regulation.

Given these findings, our analysis was geared towards regions in each gender that exhibited the most pronounced molecular changes.

#### 2.2.1. Effects of BC Treatment on Tight Junction-Related Transcripts in the Duodenum of Ztm Female Mice

Upon comparing BC-treated Ztm female mice with Ztm control female mice, several transcripts showed significant upregulation (adjusted *p*-value < 0.05). This included *Myo1c* (log2 = 20.44), *Cgn* (log2 = 20.09), *Actn1* (log2 = 21.14), *Prkacb* (log2 = 20.70), *Arhgap17* (log2 = 20.29), *Crb3* (log2 = 20.47), *Afdn* (log2 = 20.09), *Flna* (log2 = 21.82), *Cldn12* (log2 = 21.40), and *Map3k1* (log2 = 21.44).

Among these, *Cgn* (cingulin, a protein associated with cell junctions) showed a particularly marked increase in expression, with an average fragments per kilobase million (FPKM) of 6.84 in the treatment group compared to 0 in the control group and (log2 = 23.09). Furthermore, *Cldn12* (claudin-12, integral to the formation of tight junction) showed a marked increase in expression (log2 = 21.40), as shown in Table 2 and illustrated in Figure 2.

Comparing Ztm control female mice with WT reference female mice, all of the observed genes were significantly downregulated (adjusted *p*-value < 0.05) in the Ztm control group, with particularly notable decreases in *Flna* (log2 = −24.89), *Prakacb* (log2 = −23.02), and *Arhgap17* (log2 = −22.86).

When comparing the BC-treated Ztm females with the WT reference females, there was no significant change in the expression of most genes, as suggested by the q-values of one (q = 1) and FPKM of 0. This may indicate normalized expression levels of these tight-junction-related genes in the BC-treated Ztm females to the levels found in the WT reference mice.

#### 2.2.2. BC Treatment’s Influence on Tight-Junction-Related Transcripts in the Duodenum of Ztm Male Mice

When comparing BC-treated Ztm male mice to Ztm control male mice, we found significant changes in various transcripts in the duodenum (adjusted *p*-value < 0.05) associated with the tight junction. For instance, *Myh11* was significantly downregulated (log2 = −27.19), while *Arhgef2* NM_001198912 was significantly upregulated (log2 = 11.30). Similarly, *Arhgef18* was downregulated (log2 = −21.35), whereas *Ocln* was upregulated (log2 = 8.38), as shown in Table 3 and Figure 3.

In comparing control Ztm male mice with WT male mice, we observed differential expression in several genes. For example, *Myh11* was significantly upregulated (q-value = 1.06 × 10^−5^, log2 = 26.17) and *Arhgef2* NM_001198912 was significantly downregulated (q-value = 1.07 × 10^−8^; log2 = −11.58). Similarly, *Ocln* was downregulated (q-value = 4.04 × 10^−4^; log2 = −8.77).

However, when comparing BC-treated Ztm male mice with WT male control mice, *Arhgef2* was more in line with the WT reference group showing a q-value of 1, and *Ocln* showed a q-value of 1; in addition, the rest of the genes showed no expression with FPKM at 0, suggesting that BC may have normalized the gut barrier in the Ztm male mice towards the WT mice to some extent.

#### 2.2.3. BC Treatment and Its Impact on Tight Junction Transcripts in the Colon of Ztm Male Mice

Transcripts associated with tight junctions in the colon of BC-treated Ztm male mice compared with Ztm control male mice (adjusted *p*-value ≤ 0.05) were assessed and are listed in Table 4 and illustrated in Figure 4. Several transcripts were differentially expressed when comparing BC-treated Ztm males with Ztm control males. There was a notable alteration in the transcript levels of *Erbb2*, *Plec*, *Ctnnd1*, *Afdn*, and *Cldn2*; *Erbb2* showed a positive log2 value (0.62) with a statistically significant q-value (6.17 × 10^−6^), indicating an increase in its transcript, *Plec* demonstrated a substantial positive log2 value (21.07) with a q-value of 2.81 × 10^−6^, suggesting a significant increase, and *Ctnnd1* exhibited a significant rise with a log2 value of 19.79 and a q-value of 1.17 × 10^−5^.

*Afdn* showed a log2 value of 9.33 with a q-value of 0.025, suggesting a moderate increase in expression, and *Cldn2* displayed a substantial negative log2 value (−23.34) with a significant q-value (1.44 × 10^−7^), indicating a decrease in its transcript following BC treatment.

Comparing Ztm control male mice with WT reference male mice, *Erbb2* showed a negative log2 value (−0.58) with a statistically significant q-value (9.90 × 10^−^¹¹). *Plec* demonstrated a large negative log2 value (−22.00) with a significant q-value of 9.70 × 10^−7^, *Ctnnd1* showed a significant decrease with a log2 value of −20.15 and a q-value of 6.07 × 10^−6^, and *Afdn* had a negative log2 value (−9.41) with a significant q-value (0.005). Furthermore, *Cldn2* revealed a substantial positive log2 value (22.49) with a significant q-value (5.01 × 10^−7^).

However, when BC-treated Ztm male mice were compared with the WT reference male mice, all evaluated RNA transcripts (*Erbb2*, *Plec*, *Ctnnd1*, *Afdn*, and *Cldn2*) had a q-value of 1, indicating no significant differences in their expression levels between Ztm mice post BC treatment. This suggests that BC treatment might have normalized RNA expression in Ztm mice to levels similar to the WT controls.

#### 2.2.4. BC Treatment and NF-kB Pathway Modulation in Ztm Female Mice: Insights from Duodenal Transcript Analysis

In the duodenum of BC-treated Ztm female mice, several DETs associated with the NF-kB pathway were examined and compared with Ztm control female mice and WT reference female mice as shown in Table 5 and illustrated in Figure 5.

When comparing BC-treated Ztm female mice to Ztm control female mice, a significant increase in the transcripts of *App*, *Fbxw11, Nfkb1*, *Map3k7*, *Atf2*, and *Creb1* was observed. All of these showed a large positive log2 value (q-values = 0.03), indicating statistical significance.

However, when comparing Ztm control female mice with WT reference female mice, all the aforementioned transcripts had negative log2 values. *Map3k7*, *Atf2*, and *Creb1* were found to have no significant difference (q-value = 1), while the rest had q-values < 0.05, indicating a statistically significant decrease.

In the comparison between BC-treated Ztm female mice and WT reference female mice, there was no significant difference in the expression of the same genes, as all had a q-value of 1, suggesting that BC treatment may have normalized NF-kB pathway-related gene expression to WT levels in Ztm females. The log2 values showed minor fluctuations, suggesting minimal variation in gene expression between these two groups.

#### 2.2.5. Impact of BC Treatment on MyD88 Cascade Gene Expression in Ztm Male Mice Duodenum

In the duodenum of BC-treated Ztm male mice, several transcripts associated with the NF-kB signaling pathway were examined and compared with Ztm control and WT reference male mice, as shown in Table 6 and Figure 6.

When comparing expression levels between the male groups, transcripts associated with the MyD88 cascade initiated on the plasma membrane (the activation of Toll-like receptor [TLR] 4 leads to the production of inflammatory cytokines, which are signaling molecules that contribute to inflammation and immediate immune response), and the MyD88-dependent cascade initiated in endosome pathways showed significantly upregulated expression in Ztm control males compared with the Ztm control males. These genes included *Atf2* (log2 = 21.26) (MyD88-dependent cascade initiated on endosome), *Creb1* (log2 = 20.32), *Mapk7* (log2 = 20.09), and *Atf2* XM_030246840 (log2 = 20.15) (MyD88 cascade initiated on plasma membrane), as listed in Table 6 and illustrated in Figure 6.

However, comparing BC-treated Ztm male mice with Ztm control male mice, the downregulation was only significant for *Atf2* (log2 = −22.39; q-value = 2.00 × 10^−8^), *Atf2* XM_030246840 (log2 = −21.31; q-value = 2.93 × 10^−6^), and *Atf2* XM_030246844 (log2 = −20.97).

Overall, these data suggest that BC treatment in Ztm male mice may affect the expression of genes involved in the MyD88 cascade initiated on the plasma membrane and the MyD88-dependent cascade initiated in endosome pathways.

#### 2.2.6. NF-κB kB Pathway Dynamics in the Colon: Comparative Insights from BC-Treated, Ztm Control, and WT Male Mice

Analyzing the DETs, several key observations in the context of the NF-κB signaling pathway in the colon were made (Table 7 and Figure 7). In the BC-treated Ztm male mice vs. Ztm control male mice, *Nfkbiz* was downregulated (log2 fold change of −1.02) with a significant q-value (7.17 × 10^−5^), *Map2k7* was highly upregulated upon treatment with a log2 fold change of 5.02; *App* and *Cd14* transcripts showed slight upregulation and downregulation, respectively; *Dusp6* and *Ppp2r1b* were downregulated; and *Creb1* and *Atf2* showed dramatic downregulation, as seen with their massive negative fold changes.

In the Ztm controls group vs. WT reference group, *Nfkbiz* and *App* were slightly upregulated in Ztm compared to the WT, with the former being significantly so. *Map2k7* showed a notable downregulation in the Ztm control mice as opposed to the WT reference mice. *Cd14*, *Dusp6*, and *Ppp2r1b* had minimal changes when comparing Ztm control to the WT reference mice. *Creb1* and *Atf2* showed upregulation in the Ztm control mice in contrast to the WT reference mice.

For the third group, i.e., the BC-treated Ztm male mice vs. WT reference male mice, the fold change difference between treatment and WT control was minimal for *Nfkbiz*. *Map2k7* demonstrated an increase in the BC-treated Ztm group relative to the WT reference mice. *App* showed an increase, while *Cd14* and *Ppp2r1b* showed a decrease when treated Ztm were compared with WT controls. *Dusp6* experienced a more pronounced downregulation relative to WT control post-BC treatment.

From a pathway perspective, many of these genes are involved in the MyD88 cascade initiated on the plasma membrane, hinting at a potential alteration in this specific signaling cascade due to the treatment in the Ztm model. One gene, *Cd14*, is also associated with the IRAK2-mediated activation pathway. The observed changes emphasize the pivotal role of the MyD88 cascade in this context.

### 2.3. Gene Ontology Enrichment Analysis of Differentially Expressed Transcripts in Ztm Mice

To further elaborate the function of DEGs and DETs in the BC-treated Ztm mice, gene ontology (GO) annotations analysis incorporating terms correlated with the cellular component (GO_C), molecular function (GO_F), and biological process (GO_P) was carried out. There are sub-categories for each level under each major category. The GO enrichment analyses were classified and mapped.

#### 2.3.1. Female Mice—Duodenum

In the duodenum, the GO enrichment analysis of DEGs highlighted significant roles related to lipid regulation, transmembrane transport, immune response, interaction with bacteria, and specific membrane structures, as detailed in Table 8.

Significantly enriched cellular components included the apical plasma membrane, brush border membrane, extracellular space, an integral component of the presynaptic active zone membrane, and anchored component of the membrane.

The enriched molecular function observed at a significance level included insulin-like growth factor binding, phosphatidylserine binding, transmembrane transporter activity, and metallodipeptidase activity.

The significantly enriched biological processes included ion transport, antimicrobial humoral immune response mediated by an antimicrobial peptide, killing by a host of symbiont cells, regulation of presynaptic cytosolic calcium ion concentration, and response to a bacterium.

The GO enrichment analysis of DETs related to tight junctions revealed significant enrichment across multiple cellular components, molecular functions, and biological processes as depicted in Figure 8. The GO enrichment analysis highlights the various cellular components, explicitly focusing on different kinds of cell junctions and related structures.

The molecular functions predominantly revolved around protein interactions and specific binding.

The protein interactions were those central to protein binding, with subcategories emphasizing specific interactions such as actin filament binding, actin binding, and small GTPase binding. Specialized binding was highlighted by examples like potassium channel regulatory activity, LIM domain binding, and kinase-specific interactions, which include kinase binding and MAP kinase activity.

The cellular components were primarily characterized by cellular structural and interface elements such as cellular connections, with the majority defined by cellular connections and interfaces, with elements like cell junctions, bicellular tight junction, and cell–cell junctions. For the cellular framework, specific components like the apical junction complex, subapical complex, and intercalated disc were highlighted.

The biological processes provided information on cellular activities, developmental processes, and regulatory functions. Notably, there was attention to processes related to organization and assembly, such as tight junction assembly, cell–cell adhesion mediated by cadherin, and adherens junction maintenance.

Neuronal and cellular communication processes like neuron projection retraction, vesicle-mediated intercellular transport, and trans-synaptic signaling underline the importance of cellular and neural communication.

The data elucidate the intricate balance of cellular regulation and responses, exemplified by the regulation of NMDA glutamate receptor activity, behavioral fear response, and T-helper 17 type immune response. Processes like optic nerve morphogenesis and positive regulation of cell migration demonstrate the emphasis on developmental stages and cellular movement.

In summary, the terms reflect the intricate cellular components, molecular functions, and diverse biological processes that define the activities and characteristics associated with the cellular environment and junction interactions.

#### 2.3.2. Male Mice—Duodenum

In the duodenum, the GO enrichment analysis of DEGs highlighted significant roles (q ≤ 0.05) predominantly related to cell cycle regulation, DNA binding, and active cell division in the duodenal tissue, all of which are vital for maintaining the intestinal epithelial layer.

The significantly enriched cellular components include nucleus, chromosomes, nucleoplasm, and cytoplasm.

The significant molecular functions identified encompass ATP binding, DNA binding, and nucleotide binding, as detailed in Table 9.

The key biological processes observed at a significance level comprise the cell cycle, mitotic chromosome condensation, cell division, chromosome condensation, and DNA replication initiation.

The GO enrichment analysis of DETs related to tight junctions revealed significant enrichment across multiple cellular components, molecular functions, and biological processes as illustrated in Figure 9.

The cellular components chiefly concerned cellular structural features and interfaces, encompassing connections like cell junctions, cell–cell junctions, and the intricate bicellular tight junction. Specific structural elements were indicated like the Z disc, myofibril, dendritic shaft, and more pronounced components like the brush border.

The molecular functions were primarily tied to protein interactions, specifically highlighted by protein binding, with specialized forms such as actin filament binding, actin binding, and small GTPase binding. Potassium channel regulatory activity, LIM domain binding, and kinase interactions, exemplified by kinase binding and MAP kinase activity, were also highlighted.

The biological processes covered a broad spectrum of cellular activities and developmental processes, as well as organizational activities such as actin filament organization and tight junction organization. The developmental and response mechanisms, seen in processes like cellular response to mechanical stimulus, and the establishment of endothelial intestinal barriers were also highlighted. In addition, regulatory processes, in particular positive regulation of neuron migration, regulation of protein localization, and adherens junction maintenance, were also relevant.

#### 2.3.3. Male Mice—Colon

In the colon, GO enrichment analysis of DEGs identified several significantly enriched terms (q ≤ 0.05), shown in Table 10. The cellular components category comprised terms associated with various cellular components and structures. The key observations included extracellular region and space, zymogen granule membrane, and apical plasma membrane.

Within the molecular function category, several areas of significant enrichment were observed, including catalytic and transporter activities and binding. Some notable activities included hydrolase activity, serine-type endopeptidase activity, peptidase activity, sequence-specific double-stranded DNA binding, DNA-binding transcription factor activity, and vitamin D3 25-hydroxylase activity.

The biological processes category contained terms related to a variety of biological processes such as metabolic processes, localization, biological regulation, cellular processes, and response to stimuli. In particular, we noted significant enrichment in lipid metabolic processes, proteolysis, cholesterol homeostasis, response to organic substances, and response to bacterium.

The GO enrichment analysis of DETs related to tight junctions in the colon identified several key roles related to cellular structure intricacies, molecular interactions, and regulatory functions, as illustrated in Figure 10.

The cellular components showed significant enrichment in structural and interface components of the cell, such as for bicellular tight junctions, cell junctions, and cell–cell junctions, critical for communication between neighboring cells and anchoring cells in place. Hemidesmosomes and tight junctions, which are vital for connecting epithelial cells to the extracellular matrix and regulating paracellular permeability, and intermediate filaments and cytoskeletons, which provide structural support and shape to cells, were also relevant.

The molecular functions were mainly associated with structural and binding attributes, such as structural molecule activity, cytoskeletal and actin binding, and protein self-association, indicating a protein’s capability to form multimeric structures. Furthermore, JUN Kinase emphasizing key enzymatic functions in signaling pathways was also relevant.

The biological processes reflected organizational activities like actin filament and tight junction organization. Regulatory and response mechanisms such as positive regulation of neuron migration and cellular response to mechanical stimulus were also relevant.

For biological processes, terms were associated with various cellular and biological processes, ranging from cellular organization and regulation to immune response and cellular adhesion, such as bicellular tight junction assembly, actin cytoskeleton organization, and negative regulation of protein kinase activity.

In summary, the colon DEGs GO enrichment underscores a profound emphasis on cellular structural elements, molecular binding, and interactions, along with a gamut of biological processes from organization to responses.

### 2.4. KEGG and Reactome Pathway Analysis in Ztm Mice

Systematic analysis of activity was performed using KEGG (Kyoto Encyclopedia of Genes and Genomes), and genomic information was put into context with functional details. The KEGG pathway maps depict networks of interacting molecules that contribute to specific cellular functions. KEGG pathway-based enrichment analysis helps us further understand the biological functions of genes.

KEGG and reactome analysis are based on the same methodology as the GO functional enrichment analysis described above for GO enrichment. Pathways with a final q-value ≤ 0.05 were defined as those significantly enriched in differentially expressed transcripts, identifying the most essential biochemical, metabolic, and signaling pathways.

#### 2.4.1. Female Mice—Duodenumn

Specific KEGG pathways were analyzed for DETs related to tight junction with their associated term descriptions, hierarchical levels, and gene-related data. These terms represent vital pathways that play a significant role in understanding both cellular processes and disease mechanisms as shown in Figure 11.

The term “cellular community—eukaryotes” was found with 23 entries and refers to the interactions and organization processes that eukaryotic cells undergo to form complex structures, such as tissues and organs, and to ensure harmonious coexistence in multicellular organisms.

The pathway “signal transduction” which had 11 entries, describes the complex processes through which cells detect and respond to external cues. Signal transduction mechanisms convert external signals into a functional response, facilitating cellular adaptation to changing environments.

The term “infectious disease—bacterial” was found with 11 entries and delves into the molecular basis of diseases caused by bacterial infections. It provides insights into how bacteria interact with human cells and the subsequent disease mechanisms.

The term “cell motility” encompassed four entries and involves mechanisms that allow cells to move. Cell motility is crucial for numerous processes including wound healing and immune response.

“Immune system” found 10 entries and focuses on the body’s defense mechanism against foreign substances and pathogens. It provides an overview of how the immune cells detect, combat, and remember invaders to ensure the body’s safety.

The reactome analysis was conducted on DETs related to tight junctions to elucidate the underlying molecular pathways and interactions (Figure 11).

The term cell–cell communication encompassed 448 gene expressions and refers to the various interactions that enable cells to communicate with each other.

The term cell junction organization involved 352 gene expressions, describing the organization of cell junctions that are complexes that connect cells together.

EPH-ephrin-mediated repulsion of cells had 110 gene expressions. This term pertains to the signaling mechanism where EPH receptors and ephrin ligands interact, leading to cell repulsion.

The term ephrin signaling comprised 69 gene expressions and delves into the signal transduction processes that is initiated by the binding of a member of the ephrin family to its complementary receptor on the surface of an adjacent cell.

#### 2.4.2. Male Mice—Duodenum

Specific KEGG pathways were analyzed for DETs associated with the tight junction with their associated term descriptions, hierarchical levels, and gene-related data. In our exploration, several notable pathways were highlighted due to their involvement in the candidate DETs as illustrated in Figure 12.

“Cellular community—eukaryotes” had the most entries, 34; this term describes the interactions and organization of eukaryotic cells.

“Signal transduction” came next with 21 entries; this pathway elucidates how cells detect, interpret, and react to environmental signals. By transforming external cues into functional cellular responses, cells can adapt and function optimally in varying environmental conditions.

“Infectious disease—viral”: This term, with 16 entries, covers the molecular mechanisms behind diseases instigated by viral infections. It offers insights into the interactions between viruses and host cells and how these lead to disease states.

“Infectious disease—bacterial” had 15 entries and spotlights diseases stemming from bacterial infections.

The term “Immune system”, containing 14 entries, revolves around the body’s defense mechanisms.

These selected terms offer a comprehensive overview of crucial cellular activities and the molecular basis of several diseases, aiding in-depth biological understanding and potential therapeutic interventions.

The reactome analysis was conducted on DETs related to tight junctions to elucidate the underlying molecular pathways and interactions, see Figure 12. The following terms provide insights into essential cellular processes—from maintaining tissue integrity and facilitating communication to programmed cell death, emphasizing the multifaceted nature of cellular biology and research.

Three hundred fifty-two genes were associated with the term “Cell junction organization”. This is the biological process of organizing cell junctions, critical structures that ensure effective cell–cell interactions and tissue cohesion. Cell junctions are pivotal in maintaining tissue structure and facilitating cellular communication. The significant number of gene expressions signifies the intricate regulatory mechanisms governing this process.

Cell–cell communication included 448 genes. This concerns mechanisms through which cells communicate, ensuring synchronized function and intercellular coordination is fundamental for tissue functionality and homeostasis. With a substantial number of genes expressed, it underlines the process’s complexity and its essential role in various physiological activities.

Apoptosis: 460 genes were associated with this pathway. Programmed cell death is a self-regulation mechanism to eliminate damaged or unnecessary cells, ensuring tissue balance, and it is indispensable for developmental processes, immune system function, and countering harmful stimuli. The significant gene expression count reveals the myriad of regulatory pathways and factors involved in this vital cellular process.

#### 2.4.3. Male Mice—Colon

Specific KEGG pathways were analyzed from DETs associated with tight junctions in the colon, with their associated term descriptions, hierarchical levels, and gene-related data as illustrated in Figure 13. Several pathways were highlighted, such as “cellular community—eukaryotes”, and “signal transduction,” which pertains to the process by which cells interpret and respond to external signals. These processes are mediated through intricate networks of signaling molecules and receptors, ensuring the cell’s ability to react to its environment, as well as the term “Immune system”.

These terms shed light on the intricate web of cellular interactions, environmental responses, and defense mechanisms in living organisms.

The reactome analysis (Figure 13) was conducted on differentially expressed transcripts (DETs) from the colon to elucidate the underlying molecular pathways and interactions.

The biological process with the most gene expressions was “actin cytoskeleton organization”, which had 1080 gene expressions. This term is related to the organization of the actin filament structures within cells. The term “bicellular tight junction assembly” had 251 gene expressions. This process refers to the assembly of tight junctions between two cells. The term “negative regulation of protein kinase activity” encompassed 270 gene expressions. This pertains to any process that stops, prevents, or reduces the frequency, rate, or extent of protein kinase activity.

## 3. Discussion

The Ztm model, predisposed to inflammation, including neuroinflammation due to a dysbiotic microbial community and increased trafficking of pro-inflammatory bacterial species and their products into the systemic circulation, is a relevant model for studying neuroinflammatory disorders [19,21]. This novel study represents the first application of high-throughput RNA sequencing in the analysis of intestinal and brain samples from the unique Ztm mouse model.

This study aimed to assess the prophylactic application of BC on intestinal barrier function, including integrity, and to examine transcriptomic responses in the digestive tract and brain of Ztm and WT mice. Understanding the regulatory mechanisms involved in barrier function is of great importance due to the central role of maintaining barrier integrity in preventing inflammation and related diseases [1,4,6].

These findings shed light on the various regulatory mechanisms underlying the molecular interactions, functional pathways, and biological processes involved in barrier function and integrity following BC treatment.

Observing the differentially expressed transcripts, it is evident that the treatment led to varying levels of transcript expression changes across the different tissues and between the sexes. Some notable observations included the substantial upregulation in the brain of Ztm female BC when compared with WT female reference mice and the pronounced downregulation in the duodenum of Ztm male BC vs. Ztm male controls. Moreover, sex-based differences in response to treatments, as well as tissue-specific variations, are apparent in the given data.

### 3.1. Significance of Gender Effects in Bovine Colostrum Treatment for Intestinal Barrier Dysfunctions

Gender-specific differences in response to therapeutic agents, including dietary supplements like BC, are not uncommon and can be attributed to a combination of hormonal, genetic, and environmental factors [22]. In mammals, including mice, sex hormones, especially estrogens, androgens, and their receptors play a pivotal role in gut physiology [23,24]. They can influence gut motility, mucosal immune responses, and barrier functions. This might explain the differential expression of certain genes in male and female Ztm mice post-bovine colostrum treatment.

Research suggests that female and male guts might have inherent differences in microbiota composition, which could influence the local gut environment and, consequently, the response to treatment [22,25]. The differential expression of genes between female and male Ztm mice post-treatment might reflect these intrinsic differences in gut physiology and microbiota.

However, as our results have consistently shown, regardless of the gender-specific genes that were differentially expressed, the overarching outcome was a movement towards normalization of gene expression to levels seen in WT mice upon BC treatment. This indicates that while the pathways and specific genes influenced by the treatment might differ between genders, the therapeutic potential of BC remains consistent.

While our study focuses on gender-specific effects in the gut barrier of the Ztm model, it is pertinent to note that the phenomenon of sexual dimorphism in gene expression is not isolated. A comprehensive analysis involving an intercross of inbred mouse strains C57BL/6J and C3H/HeJ examined multiple somatic tissues in 334 mice. This study revealed widespread differences in gene expression between the sexes, affecting thousands of genes in liver, adipose, and muscle tissue and hundreds in the brain [26]. Such findings underline the inherent complexities and potential nuances when studying gender differences in any physiological context, including our investigation of the Ztm model.

As we delve deeper into the observed gender-specific differences in the Ztm model, it is important to recognize the multifaceted molecular mechanisms that give rise to such sexual dimorphisms. A recent comprehensive perspective elucidated the spectrum of mechanisms, from modifications to coding sequences, changes in gene regulation, to alterations during translation, that can generate sex-specific phenotypic variations [27]. Such intricacies not only emphasize the complexity inherent in sexual dimorphism but also highlight the importance of adopting advanced methodologies, as suggested by Kasimatis et al. [27]. to thoroughly understand the genotype–phenotype–fitness map.

### 3.2. Bovine-Colostrum-Mediated Restoration of Duodenal Tight Junction and Cellular Functions in Ztm Females

In BC-treated Ztm female mice, the findings show transcriptional shifts in the duodenum-associated tight junctions compared to control mice. The treatment appears to be moving the gene expression of the treated mice closer to that of the WT reference mice for the genes associated with the duodenum’s tight junction, indicating a potential normalization effect of the treatment.

In BC-treated Ztm females, a significant upregulation of several genes was observed, with the most notable being *Cgn* and *Cldn12*. Compared to WT reference females, Ztm control females showed a significant downregulation in genes like *Flna*, *Prakacb*, and *Arhgap17*.

As *Cgn* (cingulin) is involved in regulating tight junction structure, its increased expression can be seen as an attempt of the tissue to reinforce or maintain its barrier function. This can be particularly relevant in tissues where barrier function is crucial, such as the intestinal lining, to prevent unwanted molecules or pathogens from entering the bloodstream [28].

*Cldn12* (claudin-12) upregulation possibly indicates a potential strengthening or modification of the tight junctions in the duodenum. As claudins are directly involved in forming the backbone of tight junction strands, an upregulation could indicate enhanced barrier selectivity or an adaptive response to maintain homeostasis [29].

*Flna* (Filamin A) is an actin-binding protein involved in reorganizing the actin cytoskeleton. It is involved in many cellular processes, including cell migration, signaling, and maintenance of cell shape. *Prkacb* (Protein kinase cAMP-dependent type II beta) is a part of the protein kinase A (PKA) pathway, which plays a central role in mediating various cellular responses to external stimuli. *Arhgap17* (Rho GTPase-activating protein 17) is involved in the regulation of the Rho family of GTPases; in particular, it might be involved in regulating cell shape, motility, and attachment [30].

Altered cell signaling, due to changes in the PKA pathway or Rho GTPase activity, can influence various downstream processes that determine cell fate, response to external stimuli, or even metabolic shifts [31,32].

*Ephb2* (Ephb2 (EPH receptor B2) is a member of the *Eph* receptor family, which is the largest subfamily of receptor tyrosine kinases (RTKs). *Ephb2* is known to play roles in maintaining tissue boundaries. In the intestine, it is involved in maintaining the architecture of the crypt–villus axis and plays a vital role in intestinal homeostasis. Upregulation might influence cellular arrangements, tissue integrity, or signaling pathways [33].

BC-treated Ztm females appeared to have normalized expression levels of these genes, nearly matching the levels in the WT reference females.

Of particular significance is the treatment’s impact on transcriptional regulation. In a cellular context, DEGs suggest enrichment in functions related to lipid regulation, transmembrane transport, immune response, and interaction with bacteria. This may indicate better nutrient absorption, strengthening the immune defense of the compromised gut, and possibly fostering a healthier gut microbiota balance.

However, DETs heavily emphasize cellular structure and communication. The compromised gut barrier in these mice might benefit from BC, as the DETs suggest improvements in epithelial integrity.

When observing the molecular function, DEGs show functions related to binding to growth factors and lipids, which can support healing and tissue repair in the compromised gut lining. This is supported with DETs, which provide insights into protein–protein interactions that are vital for structural integrity and signaling. This might suggest a stabilizing effect on cellular structures within the gut.

Observing biological processes, DEGs emphasize the defense against microbes and nutrient transport. This means that while the gut barrier is compromised, there is an upregulated defense mechanism, possibly aided by the immunoglobulins present in the colostrum [14,15,34]. Furthermore, DETs highlight cellular assembly, adhesion, and communication, indicating a potential enhancement of gut barrier repair mechanisms, which might be supported by growth factors in colostrum [12,18,35].

The observed lipid regulation and transport suggests improved nutrient absorption. BC is known to help in the maturation of gut functions, so this could be an indication of its therapeutic effect [16].

The focus on the immune response and bacterial interactions is particularly intriguing. The compromised gut barrier can result in the translocation of bacteria and other antigens into the bloodstream, potentially leading to inflammation. BC might strengthen the gut’s defense mechanisms [36,37,38].

The neuronal and cellular communication processes suggest that, in addition to barrier repair, colostrum might also modulate neuroenteric functions, which can have implications for gut motility and signaling [39,40].

In summary, the GO enrichment analysis suggests that BC treatment might have multifaceted benefits for compromised gut barriers. These benefits range from improved nutrient absorption and enhanced immune defense to the potential repair and strengthening of the intestinal barrier.

### 3.3. Bovine-Colostrum-Induced Restoration of Gut Barrier Integrity and Cellular Dynamics in Ztm Male Mice

In the investigation of the duodenal tight junction in the BC-treated Ztm male mice, a range of significant findings emerged. Several genes were found to exhibit differential expression between the BC-treated Ztm males and the Ztm male controls and between the Ztm male controls and the WT male controls.

The gene expression data suggest that BC treatment tends to normalize gene expression in Ztm mice closer to WT levels for genes associated with tight junctions. This possibly indicates the potential therapeutic efficacy of BC in restoring gut barrier integrity in the Ztm male mice.

In both the duodenum and the colon, BC treatment appears to modulate the expression of genes associated with tight junctions such as *Ocln* and *Arhgf2* in Ztm male mice, bringing their expression patterns closer to WT mice. This could point to restoring or normalizing effect on tight junction function, which would be essential for addressing a compromised gut barrier.

Both *Ocln* (occludin) and *Arhgef2* (Rho/Rac guanine nucleotide exchange factor 2) are proteins that play key roles in cellular and tissue structures and functions. Occludin is an integral membrane protein and a component of tight junctions. In the intestinal epithelial cells, occludin helps maintain barrier function, ensuring that harmful substances in the gut lumen do not pass into the bloodstream and that necessary nutrients are absorbed efficiently.

Disruptions in occludin expression or function are implicated in various diseases, including inflammatory bowel diseases where the intestinal barrier is compromised. *Arhgef2* activates members of the Rho family of GTPases, which play fundamental roles in numerous cellular processes including cell polarity, migration, and cycle progression. *Arhgf2* expression alteration could indicate changes in cellular signaling dynamics, possibly influencing inflammation, cell migration, or tissue integrity, and has been found to be upregulated by bacterial peptidoglycan stimulation in biopsies from inflamed mucosal areas of Crohn’s disease patients [41,42].

DEGs in the duodenum were primarily associated with cell cycle regulation, DNA replication, and active cell proliferation. These findings have implications for the continuous renewal of the duodenal lining, hinting that BC may foster cellular regeneration and repair. Cellular components such as the chromosome, nucleus, nucleoplasm, and cytoplasm were featured, indicating an active role in genomic organization and cellular structure. These cellular components align with the identified biological processes, including cell cycle, mitotic chromosome condensation, and cell division.

As for molecular functions, ATP binding, nucleotide binding, and DNA binding were predominant, highlighting their essential roles in facilitating transitions within the cell cycle and stimulating DNA replication.

Moreover, the enrichment analysis of differentially expressed transcripts (DETs) further accentuates BC’s influence on structural integrity, protein interactions, and overall cellular organization, providing a comprehensive perspective on its potential benefits in the colon. DEGs highlight significant roles in metabolic processes, localization, and response to stimuli. The emphasis on lipid metabolic processes and transmembrane transport suggests BC’s effect on nutrient metabolism and transport in the colon.

Cellular component enrichments like the endoplasmic reticulum, extracellular space, and plasma membrane highlight areas of cellular activity and potential sites of action for BC.

Molecular functions underscore the diverse enzymatic and transporter activities that are being affected.

DETs primarily emphasized cellular structural components, indicating BC’s role in strengthening or restoring the structural integrity of the colon lining. The emphasis on tight junctions further supports the hypothesis of BC’s role in addressing the epithelial barrier.

In summary, BC treatment in Ztm male mice appears to play a significant role in modulating gene and transcript expression related to tight junctions and cellular structure, pointing towards its potential therapeutic role in restoring gut barrier integrity. The GO enrichment data add insights into the broader cellular and molecular processes affected by BC treatment, including cellular regeneration, structural integrity, and metabolic processes.

### 3.4. Modulatory Effects of BC Treatment on Gene Expression and Pathway Activities in Ztm Females

There was a marked increase in transcripts like *App*, *Fbxw11*, *Nfkb1*, *Map3k7*, *Atf2*, and *Creb1* in BC-treated Ztm females compared with Ztm control females. Ztm control females, when compared to WT reference females, showed a decrease in the aforementioned transcripts. Notably, *Map3k7*, *Atf2*, and *Creb1* remained statistically consistent. BC treatment seemingly normalized NF-kB pathway-related gene expressions in Ztm females to the levels in the WT reference mice.

The interpretation of the KEGG and reactome analyses from DETs in the BC-treated Ztm female mice, especially when seen in relation to DEGs and their GO terms, provided insights into the functional and pathway-based perspectives of gene expression.

In comparison with Ztm control female mice, the BC treatment led to an upregulation of genes (*App*, *Fbxw11*, *Nfkb1*, *Map3k7*, *Atf2*, and *Creb1*) in the NF-kB pathway compared with untreated Ztm female mice.

In comparison with WT reference female mice, these same genes were downregulated in untreated Ztm females when compared to WT females, suggesting a potential defect or differential pathway activity in Ztm female mice. However, with BC treatment, the pathway activity appeared to be normalized towards the WT level.

A comparison between BC-treated Ztm females and WT reference females revealed no significant difference in the expression levels of these genes, suggesting that BC treatment modulated the gene expression levels of Ztm females to be closer to those of the WT females, particularly for this pathway.

The KEGG pathway analysis showed the importance of cell–cell interactions, adherence, and tissue formation. Furthermore, signal transduction, a pathway detailing how cells perceive and respond to their environment, was highlighted. Moreover, involvement of genes in the “infectious disease—bacterial” pathway could point towards a response or susceptibility to bacterial infections or the role of gut microbiota [43,44,45].

The KEGG and reactome pathway analyses suggest a focus on cell communication, structure, and movement. These processes are vital for maintaining tissue integrity and function, especially in the gut, where tissue repair and barrier functions are vital. There is also an indication that these genes play a role in the body’s defense mechanisms, both in recognizing and responding to potential bacterial infections [46,47].

Overall, combining the KEGG, reactome, and GO analyses provides a comprehensive view of the cellular processes influenced by BC treatment in Ztm female mice. The data imply that BC treatment may restore some of the cellular functions towards a normal or WT state, especially in the context of the gut’s tissue structure, communication, and immune response.

### 3.5. Modulatory Effects of Bovine Colostrum on Immune Responses and Gut Integrity in Ztm Male Mice

Comparing the BC-treated Ztm males to Ztm controls shows significant upregulation of genes associated with the MyD88-dependent pathway, especially on the plasma membrane. BC treatment appears to reduce the expression of some of these genes, suggesting a modulatory effect on the inflammatory response.

MyD88 cascade, especially its activation on the plasma membrane, is significant in initiating an immune response. Our data suggest that BC treatment affects this cascade, potentially altering the inflammatory response in Ztm males.

There is a clear difference in gene expression profiles between BC-treated Ztm males and the control group, especially with the observed upregulation of *Map2k7* (Mitogen-Activated Protein Kinase Kinase 7 or *MKK7*) and downregulation of several genes. Map2k7 functions as a specific activator of JNK, a mitogen-activated protein kinase (MAPK) involved in the regulation of numerous physiological processes during development and in response to stress.

In the context of gut permeability and dysbiosis, the downregulation of genes in the BC-treated Ztm male group versus the Ztm male control group, especially *Cd14*, *Dusp6*, and *Nfkbiz*, suggests that BC treatment may have an impact on attenuating the inflammatory response. The IRAK2 mediated activation pathway is also emphasized, specifically by the *Cd14* gene.

*Cd14* is associated with the immune response and inflammation, playing a role in the detection of bacterial lipopolysaccharides (LPSs). Downregulation suggests a potential decrease in the inflammatory response typically associated with increased gut permeability and dysbiosis [48].

LPS recognition can lead to the activation of MAPKs, including JNK, through various signaling cascades. While *Cd14*’s primary association is with the NF-κB pathway, activation of this pathway can influence and be influenced by JNK pathways, especially during inflammatory responses.

Similarly, *Dusp6* and *Nfkbiz* are also implicated in inflammatory response pathways. *Dusp6* (Dual Specificity Phosphatase 6), also known as MKP-3 (Mitogen-Activated Protein Kinase Phosphatase 3) is a negative regulator of the MAP kinase pathway, a critical signaling pathway in inflammation, and targets the ERK pathway; there is cross-talk between the ERK and NF-κB pathways. Both pathways can be activated by similar stimuli, especially in the context of inflammation. By modulating ERK activity, it can have indirect effects on the larger MAPK signaling network, which includes JNK.

*Nfkbiz* is an ankyrin-repeat-containing protein that is induced by TNF-alpha. It is a regulator of NF-κB activation, which regulates the immune response to infection [49,50]. Under conditions of stress or infection, this pathway is activated, leading to the transcription of several inflammatory mediators, such as cytokines, chemokines, and adhesion molecules [51]. Downregulation may also point to a reduction in inflammatory response.

Like *Cd14*, while its primary role is associated with the NF-κB pathway, interactions between the NF-κB and JNK pathways, especially during inflammatory and stress responses, make it relevant when considering JNK activation.

*Atf2* (Activating transcription factor 2) and *Creb1* (cAMP responsive element binding protein 1) downregulation may be the result of the signaling pathways associated with these transcription factors being suppressed. As both can be activated by stress or inflammatory signals, this could imply a suppression of certain stress/inflammatory responses in the treated mice.

The observed changes might be indicative of a potential shift in cellular responses, influencing processes like inflammation, cell survival, or repair mechanisms in the colon.

In a broader context, the suppression of *Creb1* and *Atf2* might also influence crosstalk with the NF-κB pathway, affecting the overall inflammatory and stress response in the colon.

Furthermore, the extremely high negative fold changes for *Creb1* and *Atf2* indicate almost complete downregulation or silencing of these genes in BC-treated Ztm males [52].

When comparing Ztm control males to WT reference males, a different expression pattern emerges, indicating unique molecular signatures in these groups. Comparing the BC-treated Ztm males with WT reference males again shows varied expression patterns, reinforcing the notion that BC treatment has a significant effect on the Ztm male’s molecular profile.

Several genes were involved in the MyD88 cascade, highlighting the potential alteration in this specific signaling cascade due to BC treatment.

The data highlight the significance of the immune system pathway in both the duodenum and colon, suggesting a key role of immunity in BC treatment response in Ztm males.

In the specific circumstance of the Ztm mouse model, which exhibits a dysregulated or dysbiotic microbial community, the BC treatment revealed a decrease in microbial dysbiosis in our previous study [20]. The reduction in the dysbiotic microbial community possibly results in a diminished influx of LPS, which typically influences the TLR4 signaling pathway, consequently altering the immune responses in these mice [53,54,55,56].

The altered microbial composition generally involves decreased abundance and diversity of species and their metabolites, breakdown of the intestinal barrier integrity, and loss of goblet cells, which can result in reduced mucus secretion and thinning of the mucus layer. According to our recent findings [20]., the BC-treated Ztm mice harbored an increased abundance of beneficial species in their gut, increased eubiosis, and a reduced abundance of potentially pathogenic species.

Furthermore, BC contains several bioactive components, such as immunoglobulins, lactoferrin, and growth factors. Some of these components have been shown to have anti-inflammatory properties and could therefore impact the NF-kB pathway [12,15,57,58,59].

In summary, BC treatment in the Ztm genotype appears to downregulate critical genes involved in inflammatory and immune response pathways, potentially ameliorating the gut permeability and dysbiosis associated with the Ztm mouse model.

The tight junction-related pathways and cell–cell communication highlighted in both the KEGG and reactome analyses emphasize the importance of maintaining tissue integrity in the context of BC treatment.

The highlighted infectious disease pathways (both viral and bacterial) suggest that BC treatment might influence the host–pathogen interaction landscape, potentially affecting disease susceptibility or progression.

In conclusion, integrating these insights with the broader research context will be pivotal. These findings suggest a role of BC treatment in modulating immune response, cellular integrity, and potentially disease mechanisms in Ztm male mice. This opens avenues for understanding the exact molecular mechanisms at play and their potential therapeutic implications. However, experimental validation and further studies are necessary to establish causality and unravel the exact mechanisms of BC treatment in these models.

### 3.6. The Cerebral Landscape: Insights into the Prefrontal Cortex and Its Transcriptomic Response

Delving into the cerebral data, specifically from the prefrontal cortex, we observed limited DEGs and DETs relative to the duodenum and colon in both female and male Ztm mice. The male Ztm mice treated with BC displayed only one upregulated gene in the brain with no downregulated genes, while the BC-treated Ztm female group exhibited two upregulated and six downregulated genes in the brain. These numbers are considerably modest when juxtaposed with the duodenum and colon data.

Interestingly, despite the minimal shift in DEGs, the presence of both upregulated and downregulated DETs in the brain underscores that the effects of BC are not confined to the gut alone. This suggests a broader systemic or holistic response, with potential implications for understanding the gut–brain axis and the multifaceted influence of BC on various physiological systems.

While the cerebral data suggest a subtle BC impact, it is worth noting that the brain is a complex organ with intricate interplay between genes, signaling pathways, and cells. The minimal changes observed might indicate that the prefrontal cortex is either more resilient to BC-induced changes or that its response is more nuanced, requiring other types of analyses or experimental conditions to be fully elucidated. It is essential to understand that even though substantial shifts in gene expression were not detected in this study, even subtle cerebral transcriptomic changes can have significant implications for cognitive and neural functioning. Consequently, future investigations should delve deeper into the cerebral region, exploring various layers of the cortex, multiple cell types, and considering other research modalities, such as protein levels and post-translational modifications, to acquire a more holistic understanding of the brain’s response to BC treatment.

### 3.7. MicroRNA Signaling via Colostrum: Unraveling Post-Transcriptional Regulation and its Implications in Intestinal Development and Immunity

Alternative splicing and post-transcriptional regulation are mechanisms that contribute to the diversity of the proteome and the regulation of gene expression at the RNA level. Both mechanisms contribute to the cellular ability to produce a diverse array of proteins from a limited set of genes. While alternative splicing focuses on generating multiple mRNA variants from a single gene, post-transcriptional regulation provides a broader range of control points, ensuring that the right proteins are produced in the right amounts, at the right time, and in the right places [60].

Signaling pathways can be activated in response to external stimuli or internal cues. Dietary components can influence epigenetic modifications and cellular signaling pathways that in turn affect splicing and post-transcriptional regulation.

MicroRNAs (miRNAs) are concise, non-coding RNA fragments that play a pivotal role in post-transcriptional gene regulation. These molecules are integral to a wide array of biological and developmental functions [61].

Colostrum is rich in miRNAs with a particular emphasis on those related to immunity and development. It is posited that these miRNAs in colostrum serve as signaling agents, passed from the mother to her offspring [16]. These miRNAs are encased in extracellular vesicles, affording them protection against the aggressive environment of the digestive system. This ensures their journey to the small intestine remains unhindered, leading to their absorption and entry into the circulatory system [16].

In the realm of intestinal development, miRNAs are paramount; they promote cell health, growth, and the intrinsic stem cell activity of the intestinal lining [62]. Moreover, they play a significant role in establishing a robust immune system by modulating processes such as B- and T-cell differentiation and directing the interleukin production in macrophages [63].

Research suggests that entities like exosomes and macrovesicles may possess specific proteins that bind to the intestinal lining [64,65]. This implies that the miRNA uptake through the intestinal barrier occurs via endocytosis, wherein these vesicles disseminate their payload into the epithelial cells [65].

Treating Ztm mice with BC is essentially receiving a rich source of these miRNAs [16,66,67].

The previously discussed changes in the gene expression of Ztm mice post-BC treatment might be influenced by these miRNAs. As miRNAs are instrumental in post-transcriptional regulation, the ingestion of BC miRNAs could impact the translation of mRNAs in Ztm mice.

The emphasis on miRNAs related to immunity and development in BC could explain some of the observed changes in genes related to the MyD88 cascade and other immune response pathways. The MyD88 cascade, which was highlighted in the gene expression data, plays a central role in innate immunity [68]. It is conceivable that BC-derived miRNAs may modulate this pathway in Ztm mice, either amplifying or attenuating its activation [61,62,63].

The encapsulation of miRNAs in extracellular vesicles would facilitate the absorption of BC-derived miRNAs in the intestines of Ztm mice, allowing these molecules to exert their regulatory effects on intestinal cells or even distant tissues if they enter the circulatory system.

BC is not just a source of miRNAs, but can also contain other molecules that might affect epigenetic modifications [14,18,69]. These epigenetic changes can further influence gene expression in Ztm mice, adding another layer of complexity to the observed transcriptional alterations. This concept reinforces the idea that diet (and dietary supplements like BC) can have multifaceted impacts on the molecular biology of organisms [12,14].

In summary, the interplay between BC treatment, gene expression, miRNAs, and the broader function of the intestine is complex. It involves aspects of intestinal health and development, immune response modulation, and possibly vesicle-mediated transport of bioactive molecules [12,14,15]. This multi-faceted interaction underscores the intricate balance in the gut environment and how external factors, like BC treatment, can perturb this balance.

It needs to be emphasized that the discussion on miRNAs in BC and their potential effects on gene regulation serves as a theoretical framework based on the existing literature. While our current study did not directly assess the composition of BC, planning subsequent research to specifically investigate the presence and influence of these miRNAs, as well as other bioactive compounds, is highly relevant in relation to our murine model. As such, some aspects of this section are speculative but offer direction and insights for further exploration in this domain.

Further research is needed to detect specific miRNAs and the downstream consequences for the gut’s function and health.

### 3.8. Bovine Colostrum: A Multifaceted Agent in Gut Health, Neuroinflammation, and Beyond

From the extensive preclinical (both in vitro and animal studies) and clinical trials carried out so far, a mode of beneficial impact of BC applications has emerged [14,35,59,70]. The beneficial effect encapsulates the combined actions of its various constituents, including oligosaccharides, fatty acids, lactoferrin, lysozyme, lactoperoxidase, proline rich polypeptide (colostrinin), cytokines, chemokines, a multitude of growth factors, hormones, vitamins, and minerals. A significant number of these components exhibit synergistic interactions, i.e., amplify each other’s effects. For instance, lactoferrin, lactoperoxidase, and lysozyme collectively inhibit microbial growth more effectively than when administered individually. Similarly, the combined efficacy of lactoferrin with IgA or α-lactalbumin surpasses their individual effects [70,71,72].

BC multifaceted effects can be divided into (a) anti-microbial properties (can inflict direct harm to the cellular structures of bacteria, fungi, and parasites, and can impede viral entry into host cells and thwart their replication), (b) immune regulation (modulating the immune response, either enhancing or suppressing it, based on the physiological needs of the organism, and BC can curtail infections and mitigate inflammatory responses), (c) anti-tumor properties (directly targets tumor cells, inducing cytotoxic effects, and amplifies the body’s anti-tumor immune response), (d) metabolic regulation (oversees various metabolic processes, encompassing hematopoiesis, iron absorption, storage, glucose and lipid metabolism, and the dynamics of bone and cartilage formation and resorption, facilitates healing of skin and mucosal wounds), (e) systemic benefits (various components confer positive effects on the nervous, respiratory, gastrointestinal, and genitourinary systems), (f) digestive tract benefits (promotes the renewal and protection of the intestinal epithelium, strengthens the intestinal barrier, counteracting bacterial overgrowth and neutralizing bacterial toxins, impedes viral infections within the digestive tract, curbs inflammation, and restores balance to the intestinal microbiota), and (g) molecular mechanism (active components regulate redox processes, modulate cell receptor functions, steer cellular signaling pathways, and oversee the transcription of genes linked to pro-inflammatory and anti-inflammatory factors) [12,13,14,35,37,59,70,73,74,75,76,77,78,79,80,81,82,83,84].

In a study exploring the immunomodulatory properties of BC on cytokine production in vitro and in a novel murine–calf fecal microbiota transplantation model, the inhibition effect on IL-6 output and amplified IL-10 production was recorded. In the presence of *S. typhimurium* infection, colostrum exhibited reduced gut permeability, a decrease in immune cell penetration, and curtailed IL-6 production [36]. This emphasizes the potential of bovine colostrum in modulating the immune response and its potential therapeutic roles in addressing inflammatory conditions. Further studies are essential to understand the underlying processes and validate these outcomes in bovine models.

Animal experiments have demonstrated a prophylactic effect from BC application for reducing the disease activity of Dextran Sulfate Sodium (DSS)-induced colitis in mice when compared with control mice. The BC application decreased weight loss severity and prevented colon shortening, resulting in improved disease activity [85].

The study found that colostrum likely accelerated epithelial regeneration by stimulating cellular proliferation and differentiation, with various colostrum compounds potentially inducing functional modulation of intestinal cells [85].

Post-DSS exposure, there were alterations in the number of gamma/delta T cells, which play significant roles in regulating and resolving inflammatory processes. Colostrum pretreatment seemed to normalize these cells’ levels, indicating suppression of colitis-induced inflammation. The study also observed an elevation in myeloid-derived suppressor cells (MDSCs) post-DSS exposure, hinting at another immunoregulatory pathway in IBD pathogenesis that remains unclear [85,86].

In conclusion, this study offers a promising therapeutic approach for modulating intestinal inflammation through BC application, which induces immunoregulatory mechanisms predominantly within the innate arm of the immune system. However, further experiments are needed to understand the immunomodulatory mechanisms leading to long-term protection in IBD subjects.

Some functional properties of colostrum constituents remain unclear due to the studies being conducted in vitro. Future investigations could potentially unveil new bioactive compounds beneficial for chronic and infectious diseases.

More clinical studies, especially meta-analyses, double-blind, and placebo-controlled studies, are needed to utilize BC effectively in the food industry. Furthermore, understanding the biological roles of different BC ingredients remains a challenge for nutritionists, dieticians, researchers, and physicians.

One clinical trial found reductions in the production of IL-13 and a decrease in the frequency of CD8+ T cells expressing TNF-α in the group of children with autism spectrum disorder (ASD) after they received treatment with BC products [87].

IL-13, a cytokine typically associated with the T-helper cell subset Th2, is involved in allergic inflammation and is thought to play a role in gastrointestinal pathology commonly found in children with ASD. The study reported a reduction in IL-13 production, aligning with improved GI function and symptom reduction after the BC treatment.

TNF-α is a pro-inflammatory cytokine that can be produced by CD8+ T cells (also known as cytotoxic T lymphocytes). The study observed a decrease in the frequency of these cells expressing TNF-α after treatment with BC, indicating that pro-inflammatory signals in the gut were reduced. This decrease in inflammatory cytotoxic T lymphocytes aligns with clinical outcomes, suggesting that BC may improve gastrointestinal function by reducing the expression of pro-inflammatory cytokines in the gut.

The researchers found it interesting that previous studies showed increased TNF-α production in peripheral blood mononuclear cells isolated from children with ASD in response to specific dietary proteins, including bovine milk proteins. However, their findings suggested that the consumption of raw BC products may actually be effective for reducing pro-inflammatory cytokine production in the gut.

The present study investigating the impact of BC treatment in Ztm mice provides valuable insights into the possible prophylactic effect of BC application in addressing conditions associated with gut dysbiosis, increased gut permeability, and subsequent neuroinflammation.

Prior to the present study, we found a significant alteration in the gut microbiota of Ztm mice following BC treatment, with a shift from a dysbiotic state to a more balanced eubiotic state. This was associated with mild anxiolytic effects in both Ztm and WT mice. In the current study, transcriptomic effects were elucidated, as well as the impact on gene expression related to immune modulation, lipid metabolism, and gut-barrier function, indicating a decrease in inflammation and improved gut health.

In the context of previous studies using the DSS mouse model for IBD and studies in ASD subjects, these findings suggest a potential benefit of BC treatment not just for gut-related disorders but also for gut barrier integrity and (microbiota)-gut–brain axis disturbances where neuroinflammation plays a role.

It is important to emphasize that our findings are preliminary, and there are several steps before these results can be translated into clinical practice. The limitations of the current study are the small sample size due to few animals and no analysis of the raw colostrum material. In the future, conducting more detailed studies to understand the precise mechanisms by which BC modulates the gut microbiota and inflammation would be very beneficial. Moreover, research on how these mechanisms might be influenced by the individual’s sex, as indicated by differences between male and female Ztm mice, would also be advantageous.

Furthermore, while this study has shown promise in a specific genetically engineered mouse model, it would be essential to confirm these results in other models of disease and, ultimately, in human clinical trials. It would also be interesting to investigate whether BC treatment has long-lasting effects or whether continual treatment is needed for sustained benefit.

In conclusion, the results of this study contribute to a growing body of evidence suggesting a link between gut health and brain function and the potential for dietary interventions such as prophylactic use of BC to provide a new approach to treating a range of disorders. However, further research is needed to fully understand and exploit these potential therapeutic opportunities.

## 4. Materials and Methods

### 4.1. Animals and Experiment

The MGH Institutional Animal Care and Use Committee authorized the animal study (protocol code 2013N000013; 3 March 2021) before starting the experiment, and the guidelines for Ethical Conduct in the Care and Use of Nonhuman Animals in Research were thoroughly followed. Breeding pairs of zonulin-transgenic mouse models and a colony of C57Bl/6 wild-type mice were used in the experiment. Altogether, 12 Ztm mice (5 females, 7 males) and 13 WT mice (5 females, 8 males) were used. After weaning at four weeks, both colonies were maintained in facilities under standard conditions, i.e., 12 h light/dark cycle, standard temperature, and standard humidity. They were housed in the same facility in separate cages during the experiment and with ad libitum access to standard mouse chow and water.

Before starting the experiment, the animals were divided into four groups. Two groups (females and males) of each genotype (Ztm and WT) were fed fresh, unprocessed BC in their drinking water at a 1:1 *v*/*v* ad libitum. The remaining groups (female and male control mice) received no BC added to their drinking water. All groups received regular chow and drinking fluid ad libitum throughout the experiment. The mice were weighed thrice weekly, and their fluid intake was recorded daily.

Samples of BC were collected from authorized dairy farms in Iceland using robots, i.e., milking servers, and standard procedures were followed for the collection. The first milking BC was collected in a sterile container, filtered, refrigerated immediately at 4 °C, then transferred to one-liter sterile plastic bottles within 12 h, selected in line with regulations regarding materials and objects intended to come in contact with food. The bottles were placed immediately in a freezer at −20 °C until being exported on dry ice (−80 °C) to the Mucosal Immunology and Biology Research Center at Massachusetts General Hospital in Boston, USA. During the experiment, an a-priori-calculated amount of BC was thawed daily and mixed with drinking water provided by the animal facility in a 1:1 *v*/*v*. The feeding took place daily at the same and to secure freshness, all feeding bottles were cleaned thoroughly when the BC was replaced. The BC was well tolerated and did not induce any clinical symptoms in the animals.

After six weeks, the mice were euthanized with Isoflurane 5% with a secondary physical method of cervical dislocation.

### 4.2. Tissue Collection, Total RNA Isolation, cDNA Library Preparation, and Sequencing

The tissue samples were collected following a standard protocol. Samples from the duodenum, colon, and brain (specifically the prefrontal cortex) were immediately frozen and stored at −80 °C until further processing.

BGI Tech Solutions (Tai Po, Hong Kong) executed the total RNA extraction and RNA sequencing. Total RNA was isolated from the tissues using the TRIzol reagent (RNeasy Mini Kit, Qiagen, Hilden, Germany), adhering to the manufacturer’s instructions. An Agilent 2100 RNA Nano 6000 Assay Kit (Agilent Technologies, Santa Clara, CA, USA) was used to determine the integrity and concentration of the total RNA. Samples that had RNA integrity numbers below 7.0 were not included in subsequent analyses.

The strand-specific transcriptome library was constructed by enriching mRNA from the total RNA. Sequencing was performed on the DNBSEQ high-throughput platform and was followed by a comprehensive bioinformatics analysis. Messenger RNA (mRNA) molecules were isolated from total RNA, utilizing oligo(dT)-attached magnetic beads. These molecules were then fragmented via divalent cations in reaction systems at elevated temperatures. The first-strand cDNA synthesis utilized random N6-primed with M-MuLV Reverse Transcriptase (RNase H), which was succeeded by second-strand cDNA synthesis using DNA Polymerase I and RNase H.

Post-synthesis, the cDNA underwent end-repair and 3’ adenylation. Adapters were then ligated to the ends of these cDNA fragments. After preparing the PCR reaction system, a U-labeled second-strand template underwent UDG enzyme digestion, followed by PCR amplification. The resulting PCR products were purified using the AMPure XP system (Beckman Coulter, Carlsbad, CA, USA) and then dissolved in EB solution. The library’s integrity was validated using the Agilent Bioanalyzer 2100 system.

Following this, the double-stranded PCR products underwent heat denaturation and were subsequently circularized by a splint oligo sequence, forming the single-strand circle DNA (ssCir DNA), which constituted the final library. DNA nanoball (DNB) amplification of the library was achieved using phi29, with each DNB containing over 300 copies of a single molecule. These DNBs were loaded onto a patterned nanoarray. Single-end 50 (or paired-end 100/150) base reads were produced via combinatorial probe–anchor synthesis (cPAS).

For the subsequent analysis, SOAPnuke Version v1.5.2 (https://github.com/BGI-flexlab/SOAPnuke accessed on 22 October 2022) facilitated the filtering process. This involved the removal of reads containing adaptor pollution, reads with an N content surpassing 5%, and any low-quality reads. The remaining “Clean Reads” were saved in the FASTQ format [88].

### 4.3. Statistical Analysis

In the present study, 75 tissue samples from the duodenum, colon, and brain were sequenced using the DNBSEQ platform, generating an average of 4.50 Gb of bases per sample. The average mapping ratio to the reference genome was 97.44%, the average mapping ratio with genes was 76.94%, and 19,700 genes were identified. Hierarchical Indexing for Spliced Alignment of Transcripts v2.0.4 (HISAT) was used to align the clean reads to the reference genome. Subsequently, Bowtie [89] (v2.2.5) was applied to map the clean reads to the reference gene sequence (transcriptome). The gene expression levels were then determined using RSEM [90] utilizing the fragments per kilobase million (FPKM) metric to normalize and compare transcript levels both within and across samples. The FPKM values were transformed using the log (average + 1) method to stabilize the variance, particularly beneficial when handling data containing zeros.

### 4.4. Identification of DEGs and DETs

The R-package DESeq2 was employed, leveraging the principle of the negative binomial distribution to detect DEGs and DETs between test groups. Only DEGs/DETs with an absolute |log2 fold-change| > ±1 and an adjusted *p*-value (q-value) ≤ 0.05 were considered significant and chosen for subsequent analyses [91]. The R package *pheatmap* [92] (v1.0.12) facilitated hierarchical clustering analysis on the identified differential genes. The gene symbols and specific transcript versions for the differentially expressed genes were designated based on their accession numbers from the NCBI Reference Sequence Database (RefSeq). For context, in RefSeq, NM refers to curated mRNA sequences supported by experimental evidence, while XM denotes model mRNA sequences generated through bioinformatics methods.

### 4.5. GO, KEGG, and Reactome Enrichment Analyses

To uncover the functional roles of significant DEGs, we analyzed gene ontology (GO) [93,94] categories. Based on the Dr. Tom Data Visualization Solution System [95] of BGI Genomics, the functional enrichment of DEGs was analyzed, concentrating on the terms “biological process”, “molecular function”, and “cellular component”. All DEGs and DETs were mapped to each entry in the gene ontology database (http://www.geneontology.org/ accessed on 7 December 2022), the number of genes per entry was calculated, and a hypergeometric test was applied to find the GO function significantly enriched in candidate genes compared to all background genes of the species. The enrichment ratio is the ratio of the number of genes annotated to an entry in the selected gene set to the total number of genes annotated to the entry in the species, calculated as Rich Ratio = Term Candidate Gene Num/Term Gene Num. DEGs and DETs are expressed as q ≤ 0.05 and |log2 FC| > 1.

We performed Kyoto Encyclopedia of Genes and Genomes (KEGG) [96,97,98] pathway analyses to determine the significant pathways of the DEGs and DETs. The *p*-values were calculated based on the hypergeometric test [99] using the *phyper* package in R.

For the reactome, we used a free, open-source, curated and peer-reviewed pathway database extracted through the official mapping, NCBI2Reactome_PE_All_Levels.txt. (https://reactome.org/download-data accessed on 7 December 2022). Reactome annotation classification uses the *phyper* function in R software version 4.0.3 to perform the enrichment analysis and calculate the *p*-value, and the q-value was obtained by correction of the *p*-value. Generally, a q-value ≤ 0.05 was regarded as significant enrichment.

The *p*-value for the significance levels of terms and pathways was corrected by multiple testing, applying the q-value package in R set at a rigorous threshold (q-value ≤ 0.05) [100].

## 5. Conclusions

Our analysis underscores the impact of BC treatment on the gene expressions of tight junctions (TJs) and the Nuclear factor kappa-light-chain-enhancer of activated B cells (NF-kB) signaling pathway. This research stands as the inaugural effort in utilizing high-throughput RNA sequencing to examine intestinal and brain samples from BC-treated Ztm mice. Our data shed light on the intricate molecular interplay and pathways that govern intestinal barrier integrity when exposed to BC. By delineating the dynamics of the intestinal barrier function, this study identifies pivotal genes and their associated functional pathways in Ztm mice subjected to BC treatment.

While our findings offer a promising direction, they necessitate further exploration to both affirm these results and evaluate their relevance to dietary approaches aimed at bolstering intestinal barrier function. This fresh perspective invites deeper inquiry into specific genes and signaling pathways, exploring how these insights might translate to human health and offer potential avenues for mitigating inflammation and its associated disorders.

## Figures and Tables

**Figure 1 ijms-24-14730-f001:**
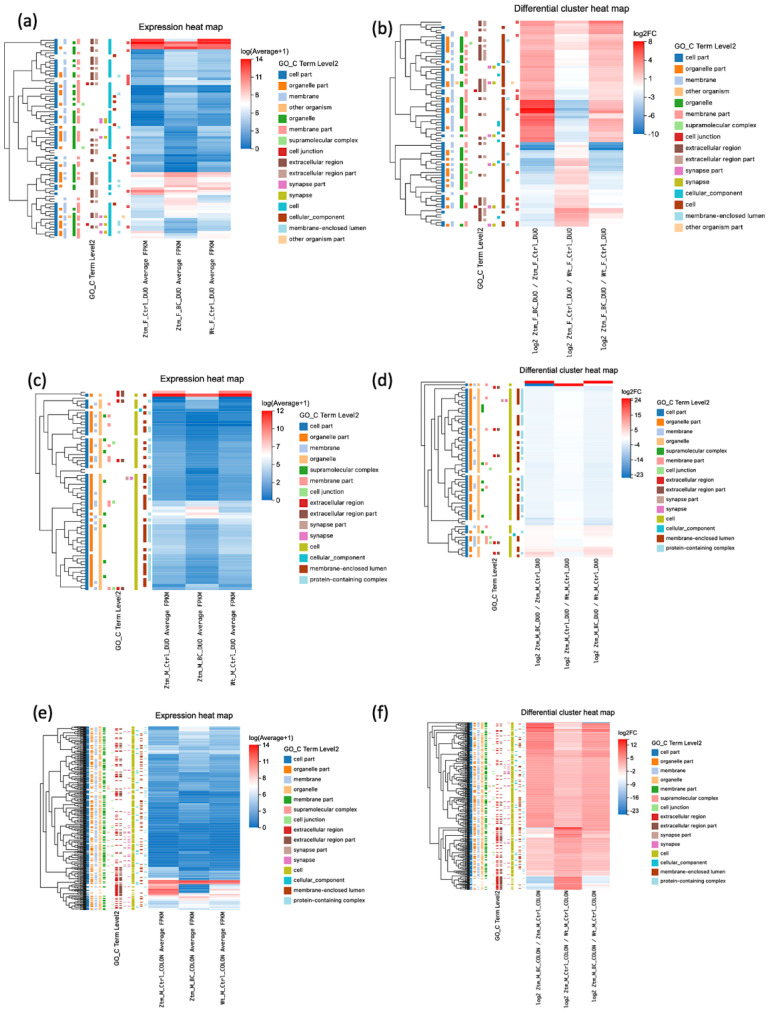
Gene expression levels across various samples and conditions. (**a**,**c**,**e**) Expression heat maps (FPKM): (**a**) female duodenum: Ztm controls, Ztm BC, WT reference mice. (**c**) Male duodenum: Ztm controls, Ztm BC, WT reference mice. (**e**) Male colon: Ztm controls, Ztm BC, WT reference mice. (**b**,**d**,**f**) Differential cluster heat maps (log2 FC) illustrating DEG expression levels between groups: (**b**) female duodenum: Comparisons of Ztm BC vs. Ztm controls; Ztm controls vs. WT reference mice; Ztm BC vs. WT reference mice. (**d**) Male duodenum: comparisons of Ztm BC vs. Ztm controls; Ztm controls vs. WT reference mice; Ztm BC vs. WT reference mice. (**f**) Male colon: comparisons of Ztm BC vs. Ztm controls; Ztm controls vs. WT reference mice; Ztm BC vs. WT reference mice.

**Figure 2 ijms-24-14730-f002:**
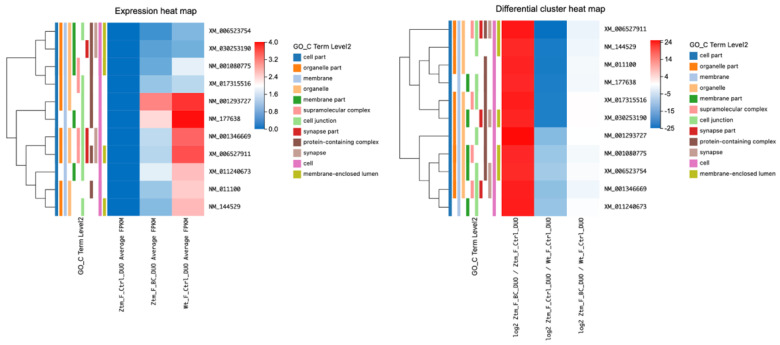
Heatmap of differentially expressed transcripts in duodenum tight junctions of Ztm female mice after BC Treatment. Expression levels are shown as average FPKM for Ztm F Ctrl, Ztm F BC, and WT F Ctrl. Differential expression is represented by Log2 FC for comparisons between Ztm F BC, Ztm F Ctrl, and WT F Ctrl. GO_C terms indicate enrichment associated with tight junctions.

**Figure 3 ijms-24-14730-f003:**
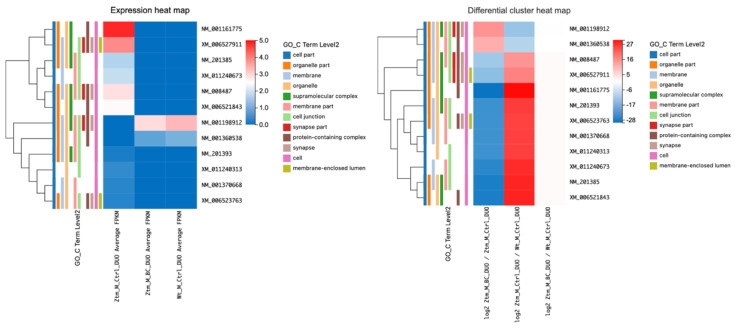
Heatmap of differentially expressed transcripts in duodenum tight junctions of Ztm male mice after BC treatment. Expression levels are shown as average FPKM for Ztm M Ctrl, Ztm M BC, and WT M Ctrl. Differential expression is represented by Log2 FC for comparisons between Ztm M BC, Ztm M Ctrl, and WT M Ctrl. GO_C terms indicate enrichment associated with tight junctions.

**Figure 4 ijms-24-14730-f004:**
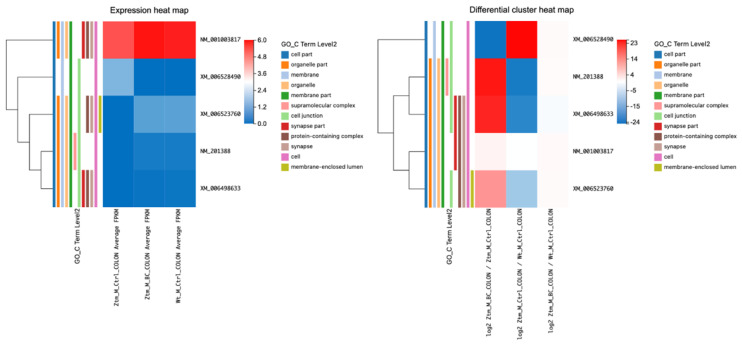
Heatmap of differentially expressed transcripts in colon tight junctions of Ztm male mice after BC treatment. Expression levels are shown as average FPKM for Ztm M Ctrl, Ztm M BC, and WT M Ctrl. Differential expression is represented by Log2 FC for comparisons between Ztm M BC, Ztm M Ctrl, and WT M Ctrl. GO_C terms indicate enrichment associated with tight junctions.

**Figure 5 ijms-24-14730-f005:**
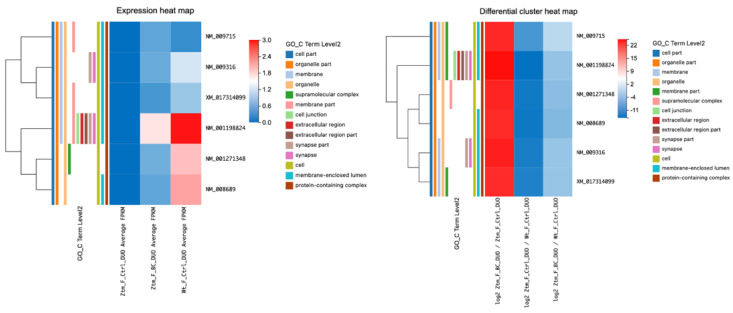
Heatmap of differential transcript expression in NF-kB signaling pathway, duodenum: BC-Treated Ztm Female Mice Comparisons. Expression represented by average FPKM for Ztm F Ctrl, Ztm F BC, WT F Ctrl. Differential expression indicated by Log2 FC for Ztm F BC vs. Ztm F Ctrl, Ztm F Ctrl vs. WT F Ctrl, and Ztm F BC vs. WT F Ctrl. GO_C terms indicate enrichment associated with NF-kB signaling pathway.

**Figure 6 ijms-24-14730-f006:**
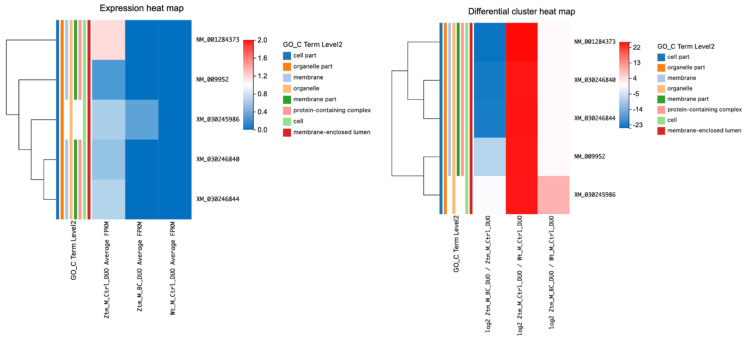
Heatmap of differential expression in the NF-kB signaling pathway in duodenum of BC-treated Ztm male mice: expression heat map: average FPKM values for Ztm M Ctrl, Ztm M BC, and WT M Ctrl. Differential cluster heat map: Log2 FC comparisons Ztm M BC vs. Ztm M Ctrl, Ztm M Ctrl vs. WT M Ctrl, and Ztm M BC vs. WT M Ctrl. GO_C terms indicate enrichment associated with NF-kB signaling pathway.

**Figure 7 ijms-24-14730-f007:**
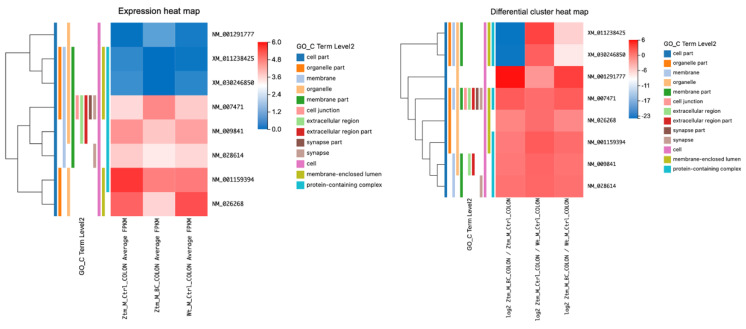
Heatmap of differential transcript expression in the NF-κB signaling pathway in colon of BC-treated Ztm male mice: expression heat map: average FPKM for Ztm M Ctrl, Ztm M BC, and WT M Ctrl. Differential cluster heat map: Log2 FC comparisons for Ztm M BC vs. Ztm M Ctrl, Ztm M Ctrl vs. WT M Ctrl, and Ztm M BC vs. WT M Ctrl. Included GO_C terms indicate enrichment associated with NF-kB signaling pathway.

**Figure 8 ijms-24-14730-f008:**
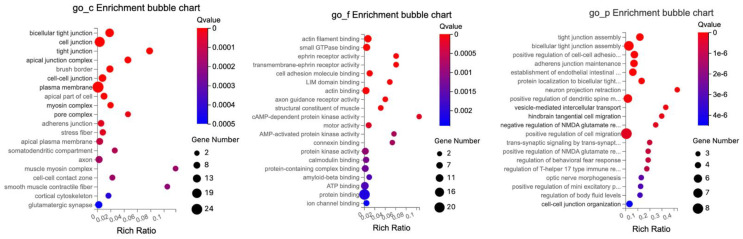
Enrichment Bubble Chart Showing Main GO Terms (CC, MF, BP) Enriched in the Duodenum (DETs) of BC-Treated Ztm Female Mice Compared with Ztm Control Female Mice.

**Figure 9 ijms-24-14730-f009:**
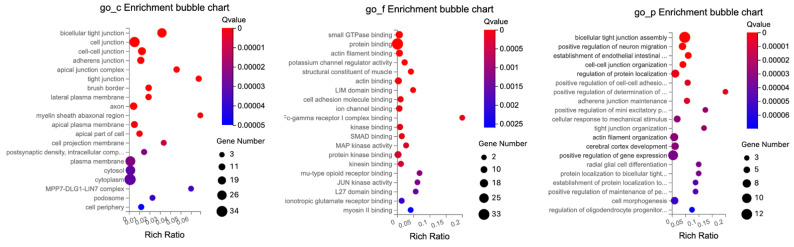
Enrichment Bubble Chart Showing Main GO (BP, CC, MF) Terms Enriched in the Duodenum (DETs) of BC-Treated Ztm Male Mice Compared with Ztm Control Male Mice.

**Figure 10 ijms-24-14730-f010:**
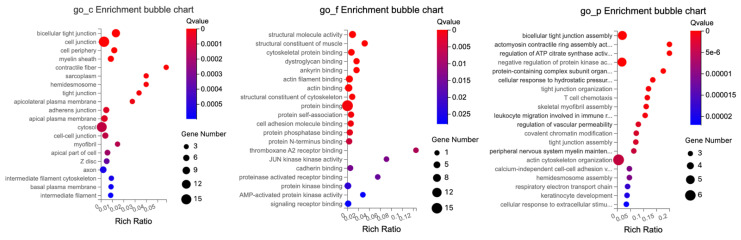
Enrichment Bubble Chart Showing Main GO (BP, CC, MF) Terms Enriched in the Colon (DETs) of BC-Treated Ztm Male Mice Compared with Ztm Control Male Mice.

**Figure 11 ijms-24-14730-f011:**
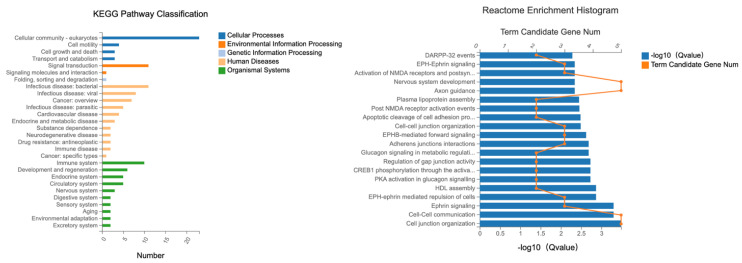
KEGG and Reactome Pathway Analyses for DETs in Duodenum of BC-Treated Ztm Female Mice: Comparisons with Ztm Control and WT Reference Female Mice.

**Figure 12 ijms-24-14730-f012:**
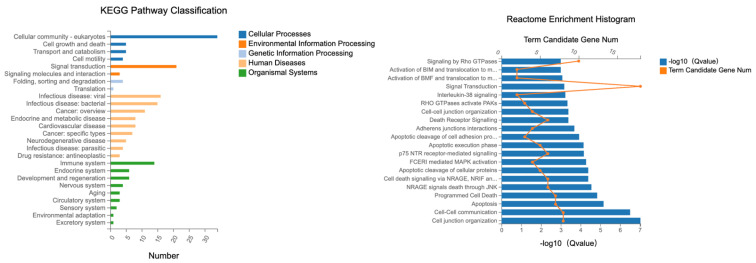
KEGG and Reactome Pathway Analyses for DETs in Duodenum of BC-Treated Ztm Male Mice: Comparisons with Ztm Control and WT Reference Male Mice.

**Figure 13 ijms-24-14730-f013:**
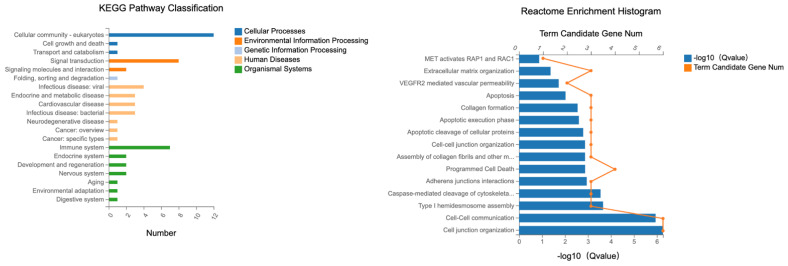
KEGG and Reactome Pathway Analyses for DETs in Colon of BC-Treated Ztm Male Mice: Comparisons with Ztm Control and WT Reference Male Mice.

**Table 1 ijms-24-14730-t001:** Differential gene and transcript expression across groups and organs. DEGs: differentially expressed genes, DETs: differentially expressed transcripts. Ztm: zonulin transgenic mice, WT: wild-type mice.

Group		Ztm Female BC vs. Ztm Female Ctrl	Ztm Male BC vs. Ztm Male Ctrl	Ztm Female Ctrl vs. WT Female Ctrl	Ztm Male Ctrl vs. WT Male Ctrl	Ztm Female BC vs. WT Female Ctrl	Ztm Male BC vs. WT Male Ctrl
Organ		Duodenum/colon/brain	Duodenum/colon/brain	Duodenum/colon/brain	Duodenum/colon/brain	Duodenum/colon/brain	Duodenum/colon/brain
DEGs	Upregulated	46/0/0	13/228/1	11/7/29	20/43/9	58/8/2	209/48/1
	Downregulated	32/4/0	54/137/6	5/3/6	8/20/1	39/11/6	193/40/0
DETs	Upregulated	1180/416/494	359/604/606	86/238/1123	982/225/370	517/575/1514	527/322/338
	Downregulated	11/76/126	1402/306/220	470/206/210	97/465/931	239/121/42	404/380/391

DEGs and DETs are expressed as q ≤ 0.05 and |log2 FC| ≥ 1.

**Table 2 ijms-24-14730-t002:** Differential Expression of Transcripts in Duodenum Tight Junctions across Ztm Female Mice Groups and WT Reference Female Mice after BC Treatment.

Duodenum—Tight Junction			Ztm F BC vs. Ztm F Ctrl		Ztm F Ctrl vs. WT F Ctrl		Ztm F BC vs. WT F Ctrl		Ztm F Ctrl	Ztm F BC	WT F Ctrl	
RNA ID	Symbol	Type	log2	q-Value	log2	q-Value	log2	q-Value	FPKM			Reactome Term Level2
NM_001080775	Myo1c	mRNA	20.44	3.20 × 10^−^^2^	−10.12	9.86 × 10^−^^3^	−1.59	1	0	0.75	2.48	Gene expression (Transcription)
NM_001293727	Cgn	mRNA	23.09	2.40 × 10^−^^2^	−12.21	2.01 × 10^−5^	−0.58	1	0	6.84	11.23	-
NM_001346669	Actn1	mRNA	21.14	3.00 × 10^−^^2^	−11.32	1.07 × 10^−^^3^	−2.07	1	0	1.79	8.29	Signaling by Rho GTPases. Miro GTPases, and RHOBTB3
NM_011100	Prkacb	mRNA	20.70	3.00 × 10^−^^2^	−23.02	8.00 × 10^−7^	−1.56	1	0	1.33	4.31	Factors involved in megakaryocyte development and platelet production
NM_144529	Arhgap17	mRNA	20.29	3.00 × 10^−^^2^	−22.86	1.18 × 10^−6^	−2.06	1	0	1.05	4.84	Signaling by Rho GTPases, Miro GTPases, and RHOBTB3
NM_177638	Crb3	mRNA	20.47	3.00 × 10^−^^2^	−22.66	1.90 × 10^−6^	−1.62	1	0	3.97	13.92	--
XM_006523754	Afdn	mRNA	20.09	3.00 × 10^−^^2^	−9.38	4.20 × 10^−^^2^	−1.21	1	0	0.39	1.00	Cell junction organization
XM_006527911	Flna	mRNA	21.82	3.00 × 10^−^^2^	−24.89	5.00 × 10^−^^3^	−2.33	1	0	1.69	9.42	Signaling by Rho GTPases, Miro GTPases, and RHOBTB3
XM_011240673	Cldn12	mRNA	21.40	3.00 × 10^−^^2^	−10.53	1.00 × 10^−^^2^	−0.85	1	0	2.45	4.72	-
XM_017315516	Map3k1	mRNA	21.44	3.00 × 10^−^^2^	−22.28	5.97 × 10^−6^	−0.30	1	0	1.28	1.74	-
XM_030253190	Ephb2	mRNA	20.68	3.00 × 10^−^^2^	−22.21	8.96 × 10^−6^	−0.34	1	0	0.63	0.86	Nervous system development

**Table 3 ijms-24-14730-t003:** Differential Expression of Tight Junction-Associated Transcripts in Duodenum of Ztm Male Mice: Comparisons between BC-treated Ztm and Ztm Controls, Ztm Controls and WT Reference, and BC-treated Ztm and WT Reference.

Duodenum—Tight Junction			Ztm M BC vs. Ztm M Ctrl		Ztm M Ctrl vs. WT M Ctrl		Ztm M BC vs. WT M Ctrl		Ztm M Ctrl	Ztm M BC	WT M Ctrl	
RNA ID	Symbol	Type	log2	q-Value	log2	q-Value	log2	q-Value	FPKM			Reactome Term Level2
NM_001161775	Myh11	mRNA	−27.19	2.28 × 10^−9^	26.17	1.06 × 10^−5^	-	-	30.14	0	0	-
NM_001198912	Arhgef2	mRNA	11.30	3.23 × 10^−15^	−11.58	1.07 × 10^−8^	−0.36	1	0	7.58	9.29	Signaling by Rho GTPases, Miro GTPases, and RHOBTB3
NM_001360538	Ocln	mRNA	8.38	3.22 × 10^−2^	−8.77	4.04 × 10^−4^	−0.47	1	0	1.07	1.42	Apoptotic execution phase
NM_001370668	Arhgef18	mRNA	−21.35	1.62 × 10^−6^	20.19	4.20 × 10^−4^	-	-	0.41	0	0	Signaling by Rho GTPases, Miro GTPases, and RHOBTB3
NM_008487	Arhgef2	mRNA	−10.85	4.13 × 10^−13^	11.02	1.86 × 10^−6^	-	-	7.41	0	0	RHOB GTPase cycle
NM_201385	Plec	mRNA	−25.09	3.46 × 10^−8^	23.23	1.14 × 10^−4^	-	-	2.97	0	0	Programmed cell death
NM_201393	Myh11	mRNA	−25.09	3.46 × 10^−8^	23.40	1.01 × 10^−4^	-	-	06.18	0	0	-
XM_006521843	Afdn	mRNA	−21.51	8.47 × 10^−7^	19.98	4.91 × 10^−4^	-	-	0.36	0	0	Cell junction organization
XM_006523763	Flna	mRNA	−12.77	3.44 × 10^−2^	12.95	3.80 × 10^−2^	-	-	14.35	0	0	Signaling by Rho GTPases, Miro GTPases, and RHOBTB3
XM_006527911	Usp53	mRNA	−21.47	9.88 × 10^−7^	20.31	4.17 × 10^−4^	-	-	0.49	0	0	-
XM_011240313	Cldn12	mRNA	−23.46	2.32 × 10^−7^	22.40	1.91 × 10^−4^	-	-	3.30	0	0	-

**Table 4 ijms-24-14730-t004:** Differential Expression of Tight Junction Transcripts in the Colon of BC-treated Ztm vs. Control and WT Reference Male Mice.

Colon—Tight Junction			Ztm M BC vs. Ztm M Ctrl		Ztm M Ctrl vs. WT M Ctrl		Ztm M BC vs. WT M Ctrl		Ztm M Ctrl	Ztm M BC	WT M Ctrl	
RNA ID	Symbol	Type	log2	q-Value	log2	q-Value	log2	q-Value	FPKM			Reactome Term Level2
NM_001003817	Erbb2	mRNA	0.62	6.17 × 10^−6^	−0.58	9.90 ×10^−^¹¹	0.04	1	52.31	88.55	83.38	Nervous system development
NM_201388	Plec	mRNA	21.07	2.81 × 10^−6^	−22.00	9.70 × 10^−7^	−0.15	1	0	0.29	0.34	Programmed cell death
XM_006498633	Ctnnd1	mRNA	19.79	1.17 × 10^−5^	−20.15	6.07 × 10^−6^	−1.20	1	0	0.09	0.18	Signaling by receptor tyrosine kinases
XM_006523760	Afdn	mRNA	9.33	2.50 × 10^−^^2^	−9.41	5.00 × 10^−^^3^	−0.08	1	0	1.72	1.87	Cell junction organization
XM_006528490	Cldn2	mRNA	−23.34	1.44 × 10^−7^	22.49	5.01 × 10^−7^	-	-	2.75	0	0	-

**Table 5 ijms-24-14730-t005:** Differential Expression in the NF-kB Signaling Pathway in Duodenum: Comparisons between BC-Treated Ztm Female Mice, Ztm Control Female Mice, and WT Reference Female Mice.

Duodenum—NF-κB kB Signaling Pathway			Ztm F BC vs. Ztm F Ctrl		Ztm F Ctrl vs. WT F Ctrl		Ztm F BC vs. WT F Ctrl		Ztm F Ctrl	Ztm F BC	WT F Ctrl	
RNA ID	Symbol	Type	log2	q-Value	log2	q-Value	log2	q-Value	FPKM			Reactome Term Level2
NM_001198824	App	mRNA	21.22	3.00 × 10^−^^2^	−10.74	1.40 × 10^−^^2^	−1.42	1	0	2.16	6.38	TRIF(TICAM1)-mediated TLR4 signaling
NM_001271348	Fbxw11	mRNA	19.74	3.00 × 10^−^^2^	−9.87	2.50 × 10^−^^2^	−2.07	1	0	0.58	2.67	Class I MHC-mediated antigen processing and presentation
NM_008689	Nfkb1	mRNA	19.44	3.00 × 10^−^^2^	−10.06	4.30 × 10^−^^2^	−2.58	1	0	0.48	3.14	MyD88 cascade initiated on plasma membrane
NM_009316	Map3k7	mRNA	20.09	3.00 × 10^−^^2^	−9.35	7.37 × 10^−^¹	−1.20	1	0	0.53	1.36	Signaling by WNT
NM_009715	Atf2	mRNA	19.45	3.00 × 10^−^^2^	−6.39	1	1.09	1	0	0.47	0.25	TRIF(TICAM1)-mediated TLR4 signaling
XM_017314099	Creb1	mRNA	19.25	3.00 × 10^−^^2^	−8.94	1	−1.27	1	0	0.33	0.88	MyD88 cascade initiated on plasma membrane

**Table 6 ijms-24-14730-t006:** Differential Expression of Duodenal Transcripts in NF-kB Signaling: Comparisons among BC-Treated Ztm Male, Ztm Control Male, and WT Reference Male Mice.

Duodenum—NF-κB Signaling Pathway			Ztm M BC vs. Ztm M Ctrl		Ztm M Ctrl vs. WT M Ctrl		Ztm M BC vs. WT M Ctrl		Ztm M Ctrl	Ztm M BC	WT M Ctrl	
RNA ID	Symbol	Type	log2	q-Value	log2	q-Value	log2	q-Value	FPKM			Reactome Term Level2
NM_001284373	Atf2	mRNA	−22.39	2.00 × 10^−8^	21.26	3.29 × 10^−4^	-	-	1.52	0	0	MyD88-dependent cascade initiated on endosome
NM_009952	Creb1	mRNA	−7.12	5.20 × 10^−^¹	20.13	4.30 × 10^−4^	-	-	0.27	0	0	MyD88 cascade initiated on plasma membrane
XM_030245986	Mapk7	mRNA	−0.92	9.70 × 10^−^¹	20.09	4.46 × 10^−4^	6.26	1	0.68	0	0	MyD88 cascade initiated on plasma membrane
XM_030246840	Atf2	mRNA	−21.31	1.93 × 10^−6^	20.15	4.26 × 10^−4^	-	-	0.63	0	0	MyD88 cascade initiated on plasma membrane
XM_030246844	Atf2	mRNA	−20.97	1.32 × 10^−6^	20.43	4.11 × 10^−4^	-	-	0.79	0	0	MyD88-dependent cascade initiated on endosome

**Table 7 ijms-24-14730-t007:** Differential Expression in the NF-kB Signaling Pathway in Colon: Comparisons among BC-treated Ztm Male Mice, Ztm Control Male Mice, and WT Reference Male Mice.

Colon—NF-κB Signaling Pathway			Ztm M BC vs. Ztm M Ctrl		Ztm M Ctrl vs. WT M Ctrl		Ztm M BC vs. WT M Ctrl		Ztm M Ctrl	Ztm M BC	WT M Ctrl	
RNA ID	Symbol	Type	log2	q-Value	log2	q-Value	log2	q-Value	FPKM			Reactome Term Level2
NM_001159394	Nfkbiz	mRNA	−1.02	7.17 × 10^−5^	0.88	1.00 × 10^−^^3^	−0.14	1	40.14	21.71	22.63	-
NM_001291777	Map2k7	mRNA	5.02	1.60 × 10^−^^2^	−2.53	1	2.47	1	0.04	1.05	0.18	MyD88 cascade initiated on plasma membrane
NM_007471	App	mRNA	0.90	2.17 × 10^−5^	−0.15	1	0.75	3.49 × 10^−5^	9.87	20.10	11.28	MyD88 cascade initiated on plasma membrane
NM_009841	Cd14	mRNA	−0.76	1.50 × 10^−^^2^	0.23	1	−0.53	7.69 × 10^−^¹	18.37	11.78	16.15	IRAK2 mediated activation of TAK1 complex upon TLR7/8 or 9 stimulation
NM_026268	Dusp6	mRNA	−1.51	3.00 × 10^−^^3^	−0.20	1	−1.72	1.16 × 10^−5^	27.87	10.46	32.89	MyD88 cascade initiated on plasma membrane
NM_028614	Ppp2r1b	mRNA	−0.50	4.20 × 10^−^^2^	0.18	1	−0.32	5.87 × 10^−^¹	11.19	8.57	10.16	Hemostasis
XM_011238425	Creb1	mRNA	−22.53	3.65 × 10^−7^	2.26	1	−5.81	9.49 × 10^−^¹	0.44	0	0.10	MyD88 cascade initiated on plasma membrane
XM_030246850	Atf2	mRNA	−22.28	4.54 × 10^−7^	0.60	1	−7.19	3.37 × 10^−^¹	0.58	0	0.42	MyD88-dependent cascade initiated on endosome

**Table 8 ijms-24-14730-t008:** Enriched GO Terms in Duodenum DEGs: BC-Treated Ztm Female Mice vs. Ztm Control Female Mice.

GO Term	GO Category	Description	Rich Ratio	q-Value
GO:0016324	Cellular Component	Apical plasma membrane	0.03	8.96 × 10^−6^
GO:0031526		Brush border membrane	0.07	4.88 × 10^−5^
GO:0005615		Extracellular space	0.01	2.80 × 10^−4^
GO:0099059		Integral component of presynaptic active zone membrane	0.11	3.63 × 10^−^^3^
GO:0031225		Anchored component of membrane	0.03	3.00 × 10^−^^3^
GO:0070273	Molecular Function	Phosphatidylinositol-4-phosphate binding	0.10	3.74 × 10^−^^3^
GO:0005520		Insulin-like growth factor binding	0.13	8.72 × 10^−^^3^
GO:0001786		Phosphatidylserine binding	0.06	9.54 × 10^−^^3^
GO:0022857		Transmembrane transporter activity	0.03	9.82 × 10^−^^3^
GO:0070573		Metallodipeptidase activity	0.29	1.00 × 10^−^^2^
GO:0006811	Biological Process	Ion transport	0.02	1.45 × 10^−^^3^
GO:0061844		Antimicrobial humoral response by antimicrobial peptide	0.05	1.45 × 10^−^^3^
GO:0051873		Killing by host of symbiont cells	0.16	5.08 × 10^−^^3^
GO:0099509		Regulation of presynaptic cytosolic calcium ion concentration	0.16	5.08 × 10^−^^3^
GO:0009617		Response to bacterium	0.03	7.73 × 10^−^^3^

Enrichment ratio is the ratio of the number of genes annotated to an entry in the selected gene set to the total number of genes annotated to the entry in the species, calculated as Rich Ratio = Term Candidate Gene Num/Term Gene Num. DEGs are expressed as q ≤ 0.05 and|log2 FC| > 1.

**Table 9 ijms-24-14730-t009:** Comprehensive Description of the Main GO Terms Enriched among DEGs in the Duodenum of BC-Treated Ztm Male Mice.

GO Term	GO Category	Description	Rich Ratio	q-Value
GO:0005694	Cellular component	Chromosome	0.05	4.67 × 10^−4^
GO:0005634		Nucleus	0.03	1.26 × 10^−8^
GO:0005654		Nucleoplasm	8.23 × 10^−^^3^	3.81 × 10^−6^
GO:0005737		Cytoplasm	5.13 × 10^−^^3^	7.04 × 10^−4^
GO:1990939	Molecular function	ATP binding	9.86 × 10^−^^3^	1.34 × 10^−^^3^
GO:0000166		Nucleotide binding	8.67 × 10^−^^3^	2.43 × 10^−^^3^
GO:0003677		DNA binding	6.89 × 10^−^^3^	3.00 × 10^−^^2^
GO:0007049	Biological process	Cell cycle	0.04	7.40 × 10^−18^
GO:0007076		Mitotic chromosome condensation	0.29	9.36 × 10^−6^
GO:0051301		Cell division	0.05	4.52 × 10^−18^
GO:0030261		Chromosome condensation	0.17	4.09 × 10^−6^
GO:0006270		DNA replication initiation	0.19	4.21 × 10^−5^

Enrichment ratio is the ratio of the number of genes annotated to an entry in the selected gene set to the total number of genes annotated to the entry in the species, calculated as Rich Ratio = Term Candidate Gene Num/Term Gene Num. DEGs are expressed as q ≤ 0.05 and |log2 FC| > 1.

**Table 10 ijms-24-14730-t010:** Primary Gene Ontology (GO) Terms Enriched in the Colon (DEGs) of BC-Treated Ztm Male Mice Compared to Ztm Control Male Mice.

GO Term	GO Category	Description	Rich Ratio	q-Value
GO:0005576	Cellular Component	Extracellular region	0.04	1.70 × 10^−9^
GO:0005615		Extracellular space	0.04	2.40 × 10^−17^
GO:0005789		Collagen containing extracellular matrix	0.05	1.12 × 10^−^^3^
GO:0016324		Apical plasma membrane	0.05	1.12 × 10^−^^3^
GO:0042589		Zymogen granule membrane	0.31	8.28 × 10^−4^
GO:0016787	Molecular Function	Hydrolase activity	0.04	1.77 × 10^−6^
GO:0008233		Peptidase activity	0.05	6.43 × 10^−5^
GO:1990837		Sequence-specific double-stranded DNA binding	0.05	6.52 × 10^−4^
GO:0003700		DNA-binding transcription factor activity	0.04	7.62 × 10^−4^
GO:0008236		Serine-type peptidase activity	0.08	1.65 × 10^−4^
GO:0030343		Vitamin D3 25-hydroxylase activity	0.33	1.00 × 10^−^^2^
GO:0006629	Biological Processes	Lipid metabolic process	0.06	2.63 × 10^−6^
GO:0006508		Proteolysis	0.05	8.31 × 10^−5^
GO:0042632		Cholesterol homeostasis	0.11	3.79 × 10^−4^
GO:0045062		Circadian regulation of gene expression	0.12	1.18 × 10^−^^3^
GO:0010033		Response to organic substance	0.09	1.18 × 10^−^^3^
GO:0014065		Response to bacterium	0.07	1.18 × 10^−^^3^

Enrichment ratio is the ratio of the number of genes annotated to an entry in the selected gene set to the total number of genes annotated to the entry in the species, calculated as Rich Ratio = Term. Candidate Gene Num/Term Gene Num. DEGs are expressed as q ≤ 0.05 and|log2 FC| > 1.

## Data Availability

Not applicable. All data generated or analyzed during this study are included in this published article.

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
