# Peer review of "Evaluating Prophylactic Effect of Bovine Colostrum on Intestinal Barrier Function in Zonulin Transgenic Mice: A Transcriptomic Study"

_ijms, 2023, doi:10.3390/ijms241914730_

Round 1

Reviewer 1 Report

Overall, this is a nicely done and well written publication. The study design is appropriate and apparently, the analyses were carefully performed. This manuscript shows rich and valuable content, which is within the journal’s scope. However, before publication some points need to be clarified.

My comments:

Line 52 – The keywords should be different from words in the title. In perspective it should increase paper's visibility. Please correct this.

Line 95 – From anatomical point of view gastrointestinal tract (also known as alimentary tract or digestive tract) is a tract which passes food from mouth to the anus. In the present study the authors studied samples of the small and large intestines only so term digestive tract is not justified.

Line 1183 – ad libitum should be written in italics.

Line 1161 – please present conclusions as a separate chapter.

Line 1200 - From histological point of view there are only four kinds of tissues: epithelial, connective, muscular and nervous. It would be better to change to “samples from the duodenum (…)” instead of “tissues from the duodenum (…)”

Line 1242 – please check the correctness of the sentence.

Reviewer 2 Report

The manuscript by Birna Asbjornsdottir and co-authors entitled: “Evaluating Prophylactic Effect of Bovine Colostrum on  Intestinal Barrier Function in Zonulin Transgenic Mice: A  Transcriptomic Study” describes results of RNA-seq analysis that was used to seek for differentially expressed genes and transcripts (DEGs and DETs, respectively) in the gastrointestinal tract and brain (prefrontal cortex) of Ztm mice and wild type mice, as well as transcripts whose expression changed after treatment of Ztm mice with bovine colostrum. The study was well-planned and well-executed; however, in my opinion too many results are presented in this manuscript, which makes it difficult to follow and interpret. The title of this article indicates that it focuses on the intestinal barrier, so the result from duodenal and colon tissues are relevant. However, additional results obtained by analyzing gene expression in the brain tissue don’t match, and are completely ignored in the Discussion section. In my opinion, the results showing DEGs and DETs in the  prefrontal cortex should be presented in a separate article focusing on the use of Ztm mouse model to study the effect of  zonulin-dependent small intestinal barrier dysfunction in relation to the brain functions. The author may create heat maps and show results of the gene ontology analysis from samples of the brain tissue in this separate article. In the present manuscript the readers may only find out about the number of DEGs and DETs in the brain of male and female mice, but the rest of the manuscript focuses on the duodenum and colon, so these results are largely ignored.

The presentation of the results of DEGs and DETs in the duodenum and colon tissues is also sometime inconsistent.

1)     Why did the authors show expression heat maps and differential cluster heat maps for duodenum (Figure 1) but didn’t show corresponding results for colon?

2)     The results clearly and consistently prove that Ztm mice treated with bovine colostrum showed gene expression similar to the wild type (healthy) mice, regardless of the gender. This confirms the beneficial effect of the bovine colostrum in promoting mucosal integrity and tissue repair. However, the authors also took into account the effect of gender in their studies. This approach is quite interesting, but was not discussed by the authors. The results for females and males are presented separately, and the discussion was divided  into subsections in which the results for females and males are discussed separately. There is no explanation or comment on why these gender-related differences occurred. When we juxtapose the result for females and males we may see that the tendencies in changes in gene expression are similar, although not the same genes are differentially expressed in both sexes. The authors should add a paragraph presenting their opinion on the significance or insignificance of the gender effect in relation to the beneficial effect of bovine colostrum in treatment of the intestinal barrier dysfunctions.

3)     The results of KEGG pathways analysis does not provide any interesting information. It is not surprising that the majority of entries was found with the “cellular community  - eukaryotes” or in very general pathways of “signal transduction”.

4)     The results of KEGG and Reactome Pathway Analyses in the colon tissue of Ztm Mice are presented in a very abbreviated way in comparison to the results of analysis in the duodenal tissue. Were the results from colon tissue less pronounced or insignificant?

5)     There is some inconsistency in presenting and numbering the subsections in the Results section:

On page 7 the authors present a subsection of the results: 2.2.2. BC Treatment's Influence on Tight Junction-Related Transcripts in the Duodenum of Ztm Male Mice.

On page 8 there is  a subsection: 2.2.3. BC Treatment and Its Impact on Tight Junction Transcripts in the Colon of Ztm Male Mice.

On page 10 there is  a subsection: 2.2.4. BC Treatment and NF-kB Pathway Modulation in Ztm Female Mice: Insights  from Duodenal Transcript Analysis.

But on page 11 there is another subsection 2.2.3. Impact of BC Treatment on MyD88 Cascade Gene Expression in Ztm Male Mice's  Duodenum;

Then on page 12 there is another subsection 2.2.4. NF-κB kB Pathway Dynamics in the Duodenum: Comparative Insights from BC-Treated, Ztm Control, and WT Male Mice.

The numbering of these subsections should be corrected. The order of the subsections should also be corrected. Why is there only one subsection about the BC treatment in female mice (on page 10) and the rest of the results are from analyses of male tissues? Why is there only one subsection about colon tissue (on page 8 and 9), and the remaining subsections describe results of analyses in the duodenum?

6)     In discussion, the paragraph describing the possible significant effect of microRNA present in the bovine colostrum on regulation of gene expression is very speculative (3.4. MicroRNA Signaling via Colostrum: Unraveling the Post-Transcriptional Regulation and its Implications in Intestinal Development and Immunity). The authors did not analyze the composition of bovine colostrum, and there are many other biologically active compounds in the colostrum that may regulate gene expression in the murine model used in this study.

Round 2

Reviewer 2 Report

The authors improved the manuscript significantly. They addressed all remarks pointed out in my review, clarifying the major and minor concerns that I had. Corrected results and information added to the discussion improved the overall quality  of the text.

Therefore, in my opinion the revised version of this manuscript should be accepted for publication.